# Notch1 signaling determines the plasticity and function of fibroblasts in diabetic wounds

Hongwei Shao[1], Yan Li[1], Irena Pastar[2], Min Xiao[3], Rochelle Prokupets[1], Sophia Liu[1], Kerstin Yu[1], Roberto I Vazquez-Padron[1], Marjana Tomic-Canic[2], Omaida C Velazquez[1], Zhao-Jun Liu[1]

Fibroblasts play a pivotal role in wound healing. However, the molecular mechanisms determining the reparative response of fibroblasts remain unknown. Here, we identify Notch1 signaling as a molecular determinant controlling the plasticity and function of fibroblasts in modulating wound healing and angiogenesis. The Notch pathway is activated in fibroblasts of diabetic wounds but not in normal skin and non-diabetic wounds. Consistently, wound healing in the $FSP-1^{+/-};ROSA^{LSL-N1IC+/+}$ mouse, in which Notch1 is activated in fibroblasts, is delayed. Increased Notch1 activity in fibroblasts suppressed their growth, migration, and differentiation into myofibroblasts. Accordingly, significantly fewer myofibroblasts and less collagen were present in granulation tissues of the $FSP-1^{+/-};ROSA^{LSL-N1IC+/+}$ mice, demonstrating that high Notch1 activity inhibits fibroblast differentiation. High Notch1 activity in fibroblasts diminished their role in modulating the angiogenic response. We also identified that IL-6 is a functional Notch1 target and involved in regulating angiogenesis. These findings suggest that Notch1 signaling determines the plasticity and function of fibroblasts in wound healing and angiogenesis, unveiling intracellular Notch1 signaling in fibroblasts as potential target for therapeutic intervention in diabetic wound healing.

## Introduction

Skin wounds heal due to a coordination of a myriad of cell types including: keratinocytes, inflammatory cells, endothelial cells (ECs), and fibroblasts (Eming et al, 2014; Ojeh et al, 2015). In recent years, there has been an increasing interest in the studies deciphering the involvement of fibroblasts in wound healing. Fibroblasts are mesenchymal cells with many vital functions, such as supporting dermal architecture, maintaining skin homeostasis under normal physiological conditions, and orchestrating the complex wound healing responses during tissue repair (Martinez-Santamaria et al,

2013; Eming et al, 2014; Liang et al, 2016; Ferrer et al, 2017; desJardins-Park et al, 2018; Stunova & Vistejnova, 2018). When tissues are injured, fibroblasts are stimulated and switch from a quiescent to activated state and transdifferentiate into myofibroblasts. These myofibroblasts which express de novo α-smooth muscle actin (α-SMA), produce abundant ECM, produce remodeling enzymes and a variety of regulatory soluble factors, provide a structural scaffold, and generate contractions to modulate and facilitate wound closure and tissue regeneration (Moulin et al, 1999; Li & Wang, 2011; Hinz, 2016; Smith, 2018). These cells are critical throughout the inflammation, proliferation, and remodeling phases of wound healing (Werner et al, 2007; Liu & Velazquez, 2008; Greaves et al, 2013; O'Brien et al, 2018; Ridiandries et al, 2018; Wallace & Bhimji, 2018). As such, (myo)fibroblasts are increasingly recognized as important therapeutic targets.

Fibroblasts display phenotypic plasticity. Fibroblast-to-myofibroblast differentiation represents a key event during wound healing. It is tightly regulated in normal wound healing but impaired in delayed or chronic non-healing wounds, which fail to progress through the orderly phases of healing and exhibit persistent inflammation, impaired angiogenesis and lack of collagen and granulation tissue in the wound bed (Falanga, 2005; Brem et al, 2008; Liang et al, 2016; Kashpur et al, 2018). Chronic non-healing wounds are often developed in patients affected by peripheral arterial disease and/or diabetes (Brem & Tomic-Canic, 2007; Eming et al, 2014; Gould et al, 2015; Pastar et al, 2018). Diabetics are also plagued by a high incidence of vascular disease that, when combined with foot ulceration, often results in lower extremity amputation (Pastar et al, 2018). For example, diabetic foot ulcers (DFUs) are one of complications of diabetes and represent a major burden on patients and the health care system. The highly inflammatory, ischemic, hypoxic, and hyperglycemic environment present in chronic diabetic wounds are generally inhibitory to myofibroblast differentiation (Tobalem et al, 2015), yet the intracellular signaling mechanisms leading to impaired fibroblast-to-myofibroblast differentiation remain largely unknown.

(Myo)fibroblasts participate in the coordinated regulation of cutaneous healing responses through an interactive dialogue with

---

[1]Department of Surgery, Miller School of Medicine, University of Miami, Coral Gables, FL, USA   [2]Department of Dermatology and Cutaneous Surgery, Wound Healing and Regenerative Medicine Research Program, Miller School of Medicine, University of Miami, Coral Gables, FL, USA   [3]Department of Surgery, School of Medicine, University of Pennsylvania, Philadelphia, PA, USA

Correspondence: zliu@med.miami.edu
Min Xiao's present address is The Wistar Institute, Philadelphia, PA, USA

their neighboring cells in the skin microenvironment, which include: inflammatory cells, keratinocytes and ECs. During inflammation, (myo)fibroblasts produce and secrete a number of cytokines and chemokines, which help to modulate the inflammatory response to injury. In the proliferative phase of wound healing, keratinocytes stimulate (myo)fibroblasts to synthesize soluble factors, which in turn stimulate keratinocyte proliferation to speed re-epithelialization in a double paracrine manner (Werner et al, 2007). Cross-talk between (myo)fibroblasts and ECs modulates wound angiogenesis, which is a critical aspect of wound healing (Velazquez et al, 2002; Li et al, 2007). The assembly of the endothelial network and stabilization of neovessels is largely dictated by external signals from (myo)fibroblasts. Newly formed blood vessels participate in the provisional granulation tissue formation and provide oxygen and nutrients to support tissue regeneration and repair. Impaired angiogenesis at the wound site is a hallmark of most chronic diabetic wounds. It is largely attributed to disrupted cross-talk between (myo)fibroblasts and ECs and insufficient support from (myo)fibroblasts.

Despite extensive evidence supporting the prominent role of (myo)fibroblasts in wound healing, intracellular signaling mechanisms that determine the reparative response of (myo)fibroblasts in wound healing, for the dysregulated fibroblast-to-myofibroblast differentiation, and impaired fibroblast-modulated angiogenesis in chronic diabetic wounds remain largely unknown. We have previously observed an inverse correlation between the statuses of Notch signaling and the activity of fibroblasts. Proliferating fibroblasts expressed either undetectable or low levels, whereas quiescent fibroblasts manifested increased levels of Notch pathway components. Consistently, loss of *Notch1* in MEFs conferred faster cell growth and motility rate, whereas constitutive activation of the Notch1 pathway slowed the cell growth and motility of human fibroblasts (Liu et al, 2012). Here, we assessed the Notch pathway activity in fibroblasts derived from human and murine diabetic wounds versus their non-diabetic counterparts and explored the role of the intracellular Notch1 pathway activity in fibroblasts in regulating wound healing and angiogenesis using novel mouse lines in which gain versus loss-of-function Notch1 signaling specifically occur in fibroblasts. We also addressed whether and how manipulation of the intracellular Notch1 pathway activity in fibroblasts alters their cross-talk with ECs by which modulate wound angiogenesis.

## Results

### Notch pathway is activated in fibroblasts of chronic diabetic wounds but not in normal skin and non-diabetic wounds

To assess the status of Notch signaling in fibroblasts of normal skin versus chronic non-healing DFU tissues, we generated primary fibroblasts from healthy foot skin of non-diabetic donors and non-healing DFU of patients and compared their Notch pathway activity. Primary fibroblasts were generated from DFU at the site of wound edge from three patients and normal foot skin specimens of three non-diabetic donors and characterized as described previously (Liang et al, 2016; Jozic et al, 2017). These primary cells were named diabetic foot ulcer fibroblast (DFUF) and NFF, respectively. Levels of

expression of Notch pathway components (Notch receptors, ligands, and targets) was determined by immunoblotting. The levels of Notch1-4, Jagged 1-2, Delta-like (Dll) 1, 3, 4, Hes-1, and Hey-1 were significantly higher in DFUF than NFF (Fig 1A). NFF expressed marginal or undetectable levels of Notch 2-4, Jagged 1-2, Dll 4, and two targets (Hes-1 and Hey-1). Two NFF expressed basal levels of Notch1, but active form of Notch1 was undetectable. These data indicated activation of the Notch pathway in DFUF, whereas it is inactivated in NFF. It is unclear which Notch ligand is primarily responsible for the Notch1 activation in DFUF. Inhibition of ligand-induced Notch activation by DAPT, a γ-secretase inhibitor, could significantly inhibit the Notch1 pathway activation by reducing the levels of Notch1 and Hey-1 in DFUF (mixture of three DFUF at 1:1:1 ratio), whereas Jagged-1–neutralizing antibody achieved a less extent inhibition compared with DAPT (Fig 1B). These results suggest that intracellular Notch pathway activation observed in DFUF is dependent upon Notch receptor–ligand interaction. Likely, all ligands contribute to the Notch pathway activation, as blocking of a single type of Notch ligand (by Jagged-1 neutralizing antibody) only partially suppress the Notch pathway activation.

The Notch pathway activity in fibroblasts of normal foot skin versus DFU was also examined in human tissue specimens by immunostaining. Fibroblasts in papillary and reticular layers of dermis were stained with antibody against FSP-1 and the Notch pathway activity was assessed by Hes-1 and Hey-1 (Notch targets) expression. Compared with fibroblasts located both in papillary and reticular layers of dermis in control foot skin, which exhibited barely detectable levels of Hes-1, expression of Hes-1 was higher in fibroblasts (Hes-1 is located in the nucleus [pink color] and cytoplasm [orange color]) at DFU tissue (Fig 1C). The combination of three colors (Hes-1, FSP-1 [fibroblast-specific protein-1, also known as S100A4], and DAPI) are shown in Fig 1B. Fibroblasts in reticular layers of dermis are framed with dash lines. See Fig S1 for images and the ratio of Hes-1:FSP-1 in highlighted reticular layers, and Fig S2 for images the ratio of Hes-1:FSP-1 in highlighted papillary layers. Similar pattern of Hey-1 in fibroblasts presented in reticular layers of dermis was observed (Fig S3). These results confirm the activation of Notch pathway in fibroblasts from and at DFU.

In addition, we also assessed the Notch pathway activity in fibroblasts of murine normal skin wounds versus ischemic (non-diabetic) skin wounds versus diabetic skin wounds by immunostaining. Skin wounds were created on dorsal skin of C57 BL6 (normal non-ischemic acute wounds), ischemic limb skin of C57 BL6 (non-diabetic ischemic chronic wounds), NOD (type I diabetes), and db/db (type II diabetes) mice by punch biopsies. Wound tissues were harvested at day 1 (early time point), day 7 (middle time point) post-wounding in all mice, and at day 9/13/14/15 (late time point, when wounds are healed) in C57 BL6 (normal non-ischemic acute wounds)/C57 BL6 (non-diabetic ischemic chronic wounds)/db/db/NOD mice, respectively. Wound tissues are subjected to immunostaining with anti–FSP-1 and anti–Hes-1 and anti-N1IC antibodies. Similarly, expression of Hes-1 was higher in fibroblasts at diabetic wounds, both NOD and db/db mice, than that in normal and ischemic wounds in C57 BL6 mice at day 7 (Fig 1D). See Fig S4 for images with individual color and the ratio of Hes-1:FSP-1 in normal or ischemic skin wound (C57 BL6 mice) and diabetic skin wound (NOD and db/db mice). Levels of Hes-1 in fibroblasts presented in

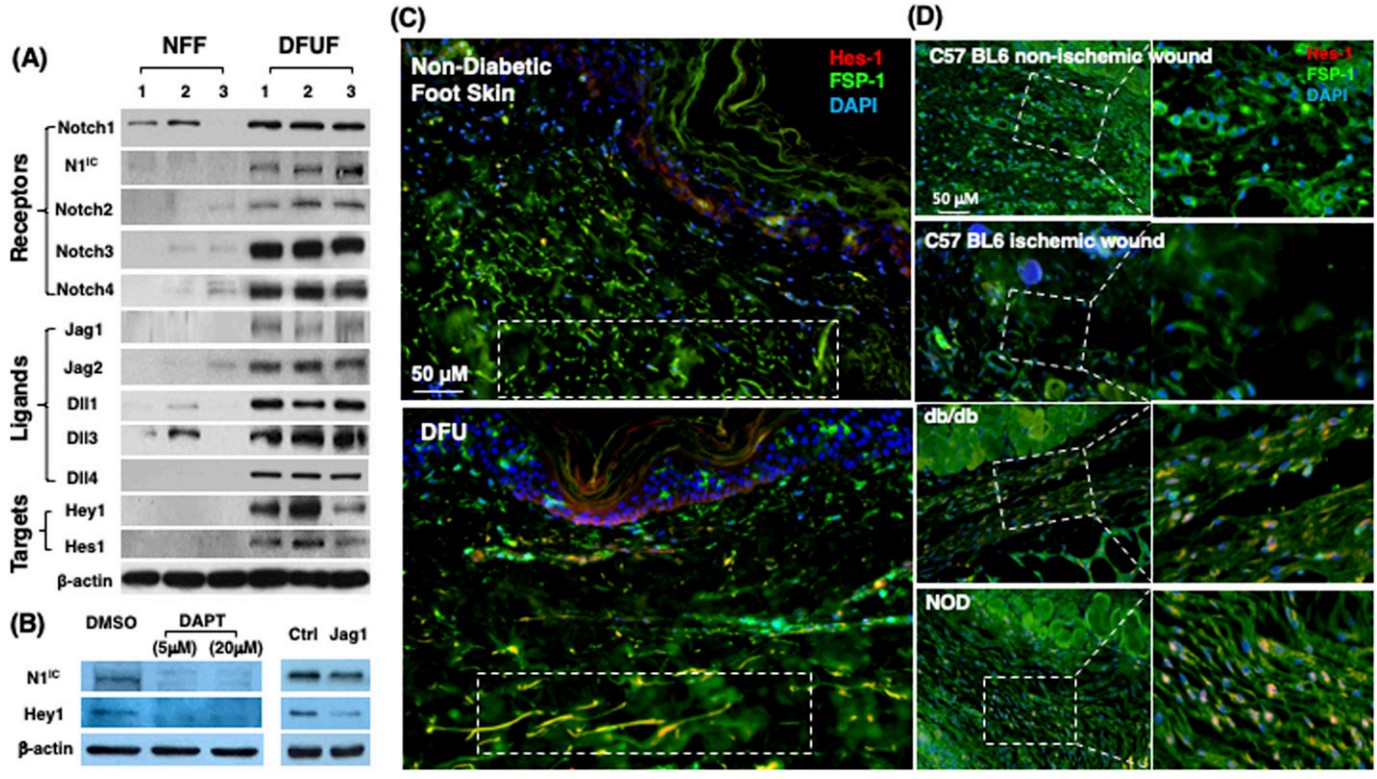

**Figure 1.  Differential Notch pathway activities in fibroblasts of chronic diabetic skin wounds versus non-diabetic skin and wounds.**
**(A)** High Notch pathway activity in diabetic foot ulcer fibroblasts (DFUF) versus low Notch pathway activity in normal foot fibroblasts (NFF). Expression of Notch pathway components in three DFUF and three NFF were assessed by immunoblot. $\beta$-actin was used as a loading control. The band of each molecule is shown. **(B)** Inhibition of the Notch pathway activity, reflected by decreased levels of N1$^{IC}$ and Hey-1, in DFUF by DAPT and Jag 1 neutralizing Ab. Compared with DAPT, Jag 1 neutralizing Ab only achieved a partial inhibition. **(C)** Representative immunostaining images show that fibroblasts (green) express higher levels of Hes-1 (red) in skin at the edge of diabetic foot ulcer tissue than that in non-diabetic foot skin. Highlighted areas show fibroblasts in reticular layers. **(D)** Representative immunostaining images show that fibroblasts (green) express higher levels of Hes-1 (red) in wounds of diabetic mice (db/db and NOD) but not in non-diabetic acute wound and ischemic chronic wounds in C57 BL6 mice. Wound tissues were harvested at day 7. Highlighted areas show fibroblasts in granulation tissues.

diabetic wounds (NOD and db/db mice) were detectable at day 1, peaked in day 7, and reduced to a very low level when the wounds were healed in diabetic mice (NOD and db/db), but remained undetectable throughout wound healing process in both normal acute and non-diabetic chronic wounds (C57 BL6 mice) (Fig S5). These results not only confirm that the Notch pathway is activated or turned "ON" in fibroblasts in murine diabetic wounds, whereas inactivated or turned "OFF" in fibroblasts in normal and non-diabetic murine wounds but also show dynamic changes of the Notch pathway activity over the wound healing process. Similar results of N1$^{IC}$ levels in various types of wounds are shown in Fig S6. Taken together, our data demonstrated that the Notch pathway is activated or turned "ON" in fibroblasts from human DFU and murine diabetic wounds, whereas inactivated or turned "OFF" in fibroblasts derived from normal (both human and murine) or non-diabetic murine skin wounds.

### Activation of Notch1 in fibroblasts delayed skin wound healing in mouse model

To address whether the activation of Notch1 in fibroblasts affects skin wound healing, we generated and tested two mouse lines, in which activation or inactivation of Notch1 pathway specifically occurs in fibroblasts. Expression of N1$^{IC}$ (Notch1 intracellular domain, an active form of Notch1) in skin fibroblasts of Gain-Of-Function Notch1 (GOF$^{Notch1}$: $Fsp1.Cre^{+/-};ROSA^{LSL-N1IC+/+}$) and deletion of Notch1 in skin fibroblasts of Loss-Of-Function Notch1 (LOF$^{Notch1}$: $Fsp1.Cre^{+/-};Notch1^{LoxP/LoxP+/+}$) mice were validated by immunostaining (Fig S7). GOF$^{Ctrl}$ ($FSP1.Cre^{-/-};ROSA^{LSL-N1IC+/+}$) and LOF$^{Ctrl}$ ($FSP1.Cre^{-/-};Notch1^{LoxP/LoxP+/+}$) mice were used as corresponding control (Shao et al, 2015). Excisional wounds were created on the dorsal skin of GOF$^{Notch1}$ versus GOF$^{Ctrl}$ mice and LOF$^{Notch1}$ versus LOF$^{Ctrl}$ mice by 6-mm punch biopsies (n = 6/group). All mice exhibited normal skin structure, cellular morphology, and no lymphocyte infiltration (data from GOF$^{Notch1}$ versus GOF$^{ctrl}$ mice are shown in Fig S8). Similar data are obtained in LOF$^{Notch1}$ versus LOF$^{ctrl}$ mice (data not shown) in skins of 4-wk-old mice, except less collagen deposition in the skin of GOF$^{Notch1}$ mice as examined by Masson's trichrome staining (data from GOF$^{Notch1}$ versus GOF$^{ctrl}$ mice are shown in Fig S9), which is consistent with impaired function of fibroblasts because of increased intracellular Notch1 pathway activity. Wound healing rates were measured by daily digital photography and wound closure was measured using ImageJ. We found that the skin wound healing of GOF$^{Notch1}$ mice was

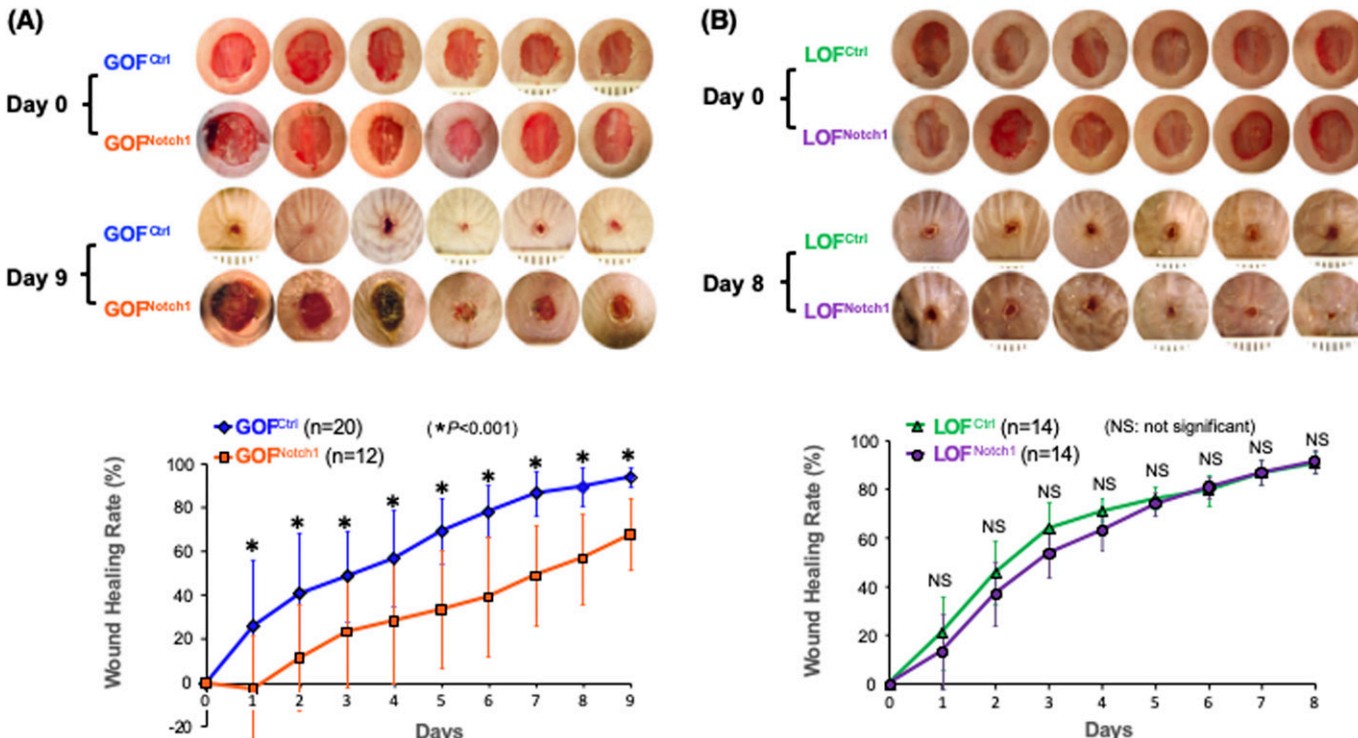

**Figure 2. Activation of Notch1 pathway in fibroblasts delays skin wound healing in mouse models.**
**(A)** Mouse skin wound healing was delayed in GOF[Notch1] mice compared with GOF[Ctrl] mice. *Top*: six representative images of wounds in each group at Day 0 and Day 9 were shown. *Bottom*: Wound healing curves. Numbers of mice in each group is listed. **(B)** Mouse skin wound healing rates were comparable between LOF[Notch1] and LOF[Ctrl] mice. *Top*: six representative images of wounds in each group at Day 0 and Day 8 were shown. *Bottom*: Wound healing curves. Numbers of mice in each group is listed. All data are analyzed by two way ANOVA followed by post-hoc tests and presented as percentage wound closure (recovery), mean ± SD from each group.

significantly delayed compared with GOF[Ctrl] mice (Fig 2A). At day 9 post-wounding, 94% of wound areas were healed in GOF[Ctrl] mice, yet only 67.9% of wound areas were covered in GOF[Ctrl] mice ($P <$ 0.01). Wound healing rates between LOF[Notch1] versus LOF[Ctrl] were comparable (Fig 2B). Overall, these results revealed that similar to DFU patients, the activation of Notch1 pathway targeted to dermal fibroblasts (DFs) delayed wound healing in the mouse model.

### Activation of Notch1 pathway suppresses cellular proliferation and migration of fibroblasts

We further tested the effect of the Notch1 pathway activation on cell proliferation and migration of DFs. For this purpose, DFs were isolated from GOF[Notch1] versus GOF[Ctrl] mice and LOF[Notch1] versus LOF[Ctrl] mice using standard methods (Shao et al, 2015). The generated mouse DFs were characterized (Fig S10) and named GOF[Notch1]-derived fibroblasts (GOF[Notch1]-DF), GOF[Ctrl]-DF, LOF[Notch1]-DF, and LOF[Ctrl]-DF accordingly. We carried out WST cell proliferation assay to assess cell growth of these DF and found that high intracellular Notch activity significantly retarded the cell growth of DFs. Cell proliferative rate of GOF[Notch1]-DF was ~50% slower than that of GOF[Ctrl]-DF (Fig 3A). Consistently, cellular proliferative activity of fibroblasts in the wound environment of mice was fairly low in GOF[Notch1] mice compared with that in GOF[Ctrl] mice. No obvious difference between the LOF[Notch1] and LOF[Ctrl] mice was found, as evidenced by a decreased expression of the cellular proliferation

marker Ki67 in fibroblasts at wound tissues of mice detected by immunostaining (Fig 3B). We also conducted an in vitro wound healing assay to test effect of the Notch1 pathway activation on cell migration of DF. GOF[Notch1]-DF versus GOF[Ctrl]-DF and LOF[Notch1]-DF versus LOF[Ctrl]-DF were growing in 24-well plates to reach confluence and formed monolayers. A cell-free pseudo-wound field (500 $\mu$m diameter) was then created in the center of the well. Cells were visualized at various time points post-"wounding" under the microscope at 4× magnification and images were acquired. Cell-free pseudo-wound fields covered by cells were calculated as percentage using ImageJ. We observed that migration and proliferation of GOF[Notch1]-DF was significantly slower than GOF[Ctrl]-DF. At 40-h post-"wounding," 95.4% of "wound" areas were covered by GOF[Ctrl]-DF, compared with only 48.3% of wound coverage by GOF[Notch1]-DF ($P < 0.05$) (Fig 3C). No significant difference between LOF[Notch1]-DF and LOF[Ctrl]-DF was observed, which is consistent with wound healing experiments. These results showed that high intracellular Notch activity suppressed cellular proliferation and migration of fibroblasts.

### High intracellular Notch1 activity inhibits differentiation of fibroblasts into myofibroblasts

In the analysis of the collagen content in the wounds, we found a decrease of collagen deposition in the wound tissues of GOF[Notch1] mice compared with GOF[Ctrl] mice, although no significant difference in the wounds of LOF[Notch1] versus LOF[Ctrl] mice was found (Fig 4A). This finding suggests that the composition of myofibroblasts in

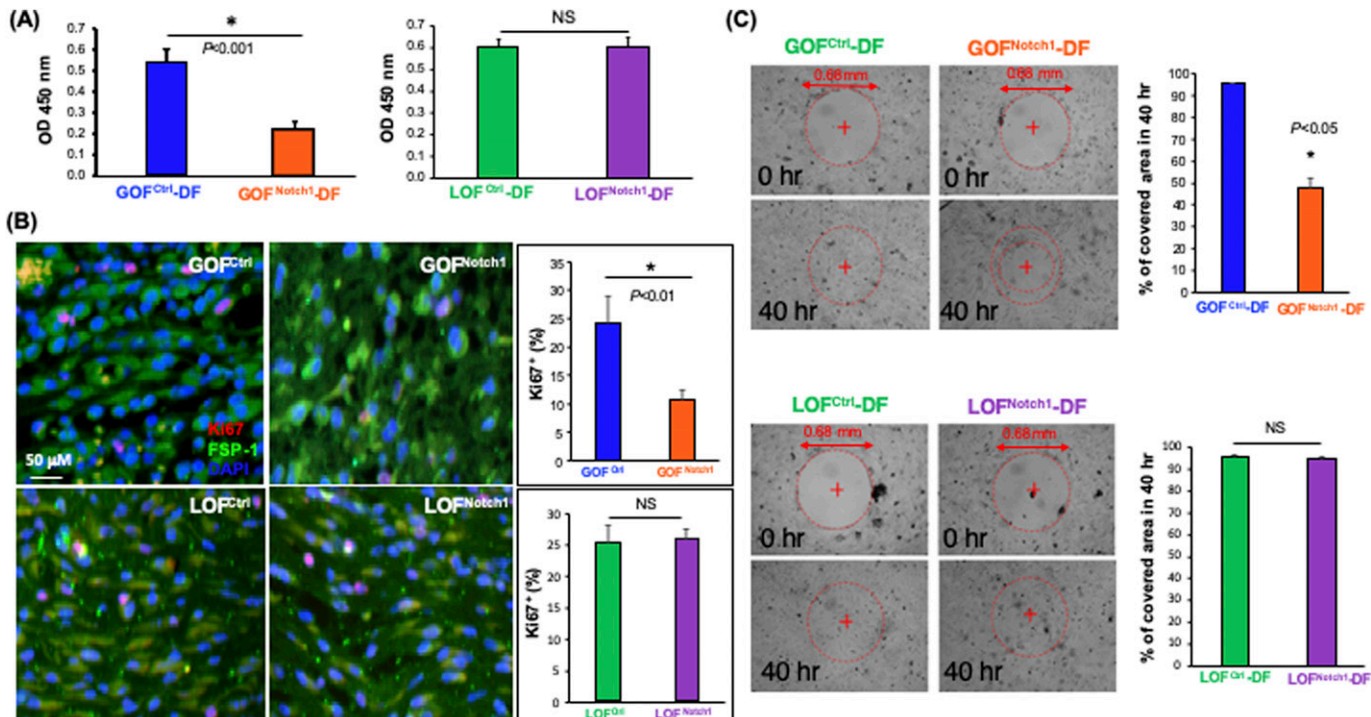

**Figure 3. Activation of Notch1 pathway in fibroblasts suppresses cellular proliferation and migration.**
**(A)** GOF[Notch1]-(DF) dermal fibroblast grew slower than GOF[Ctrl]-DF, whereas growth rates of LOF[Notch1]-DF and LOF[Ctrl]-DF were comparable. Data of mean ± SD are based on results of three experiments of total six wells/group (cells grew in 96-well plate) and analyzed by $t$ test. **(B)** Expression of cell proliferation marker Ki67 (red) is lower in fibroblasts (FSP-1, green) at wound granulation tissue of GOF[Notch1] mice than that in GOF[Ctrl] mice, yet no obvious difference between LOF[Notch1] and LOF[Ctrl] mice. Quantitative data are calculated based on three sections/wound and mean ± SD are analyzed by $t$ test. **(C)** GOF[Notch1]-DF migrate and proliferate were slower than those from GOF[Ctrl]-DF, whereas migration and proliferation of LOF[Notch1]-DF and GOF[Ctrl]-DF were comparable as assessed by in vitro wound healing assay. Data of mean ± SD are based on results of three experiments of total six pseudo-wounds/group and analyzed by $t$ test.

wound tissues of GOF[Notch1] mice may be dysregulated because myofibroblasts are the major source of newly made collagen during wound healing. To explore the mechanisms for delayed wound healing and decreased collagen deposition in the GOF[Notch1] mouse, we investigated whether Notch1 activation in fibroblasts affects cellular plasticity and differentiation into myofibroblasts. Myofibroblasts can be characterized by the neoexpression of α-SMA, the active production of collagen, the presence of several remodeling enzymes, and contraction of collagen gel in vitro (Darby et al, 1990; Kissin et al, 2006; Minz et al, 2010). Thus, we tested expression levels of α-SMA in these DFs cultured with complete DMEM in vitro. Under regular culture conditions, fibroblasts can be activated by serum and cytokines contained in culture media to gain phenotypic characteristics of myofibroblasts. As expected, all DF expressed high levels of α-SMA except GOF[Notch1]-DF, which carry high Notch1 activity (Fig 4B). GOF[Notch1]-DF also displayed less microfilaments. Consistently, GOF[Notch1]-DF displayed weak cellular capability to contract collagen gel in vitro compared with GOF[Ctrl] as shown in fibroblast-mediated 3D type I collagen gel assay (Fig 4C). In all assays, we did not observe any significant difference between LOF[Notch1]-DF and LOF[Ctrl]-DF. This is consistent with unaltered wound healing rates between LOF[Notch1] and LOF[Ctrl] mice as shown above.

Moreover, we examined the levels of α-SMA in myofibroblasts at wound granulation tissues of GOF[Notch1] versus GOF[Ctrl] and LOF[Notch1] versus LOF[Ctrl] mice by immunostaining. As shown in Fig 4D, myofibroblasts in

wounds of GOF[Notch1] mice expressed lower amounts of α-SMA than that in GOF[Ctrl] mice. These in vivo results are consistent with the results of in vitro assay as shown in Fig 4B despite that myofibroblasts in the wound did not spread and stretch as well as in vitro. Taken together, our in vitro and in vivo data indicate that high intracellular Notch1 pathway activity inhibits the plasticity of fibroblasts and blocks the differentiation of fibroblasts into myofibroblasts. In turn, impaired cellular plasticity of fibroblasts and decreased myofibroblasts result in delayed wound healing.

## Activation of Notch1 pathway in fibroblasts mitigates wound angiogenesis

Another important component of skin wound healing is the formation of new blood vessels in granulation tissues to provide nutrients and oxygen to support tissue repair. Therefore, we examined the potential effect of the Notch1 pathway activation in fibroblasts on the neovascularization in wound tissues. We conducted a live animal whole body perfusion using a formulated aqueous solution containing DiI and followed by scanning the entire wound tissue using laser scanning confocal microscopy to visualize the vascular network in the wound. We observed a significant lack of neovascularization in the wounds of GOF[Notch1] mice when compared with the wounds of GOF[Ctrl] mice (Fig 5A). Typically, neovascularization develops and sprouts from the edge of wounds and moves towards the center of the wound bed. More mature

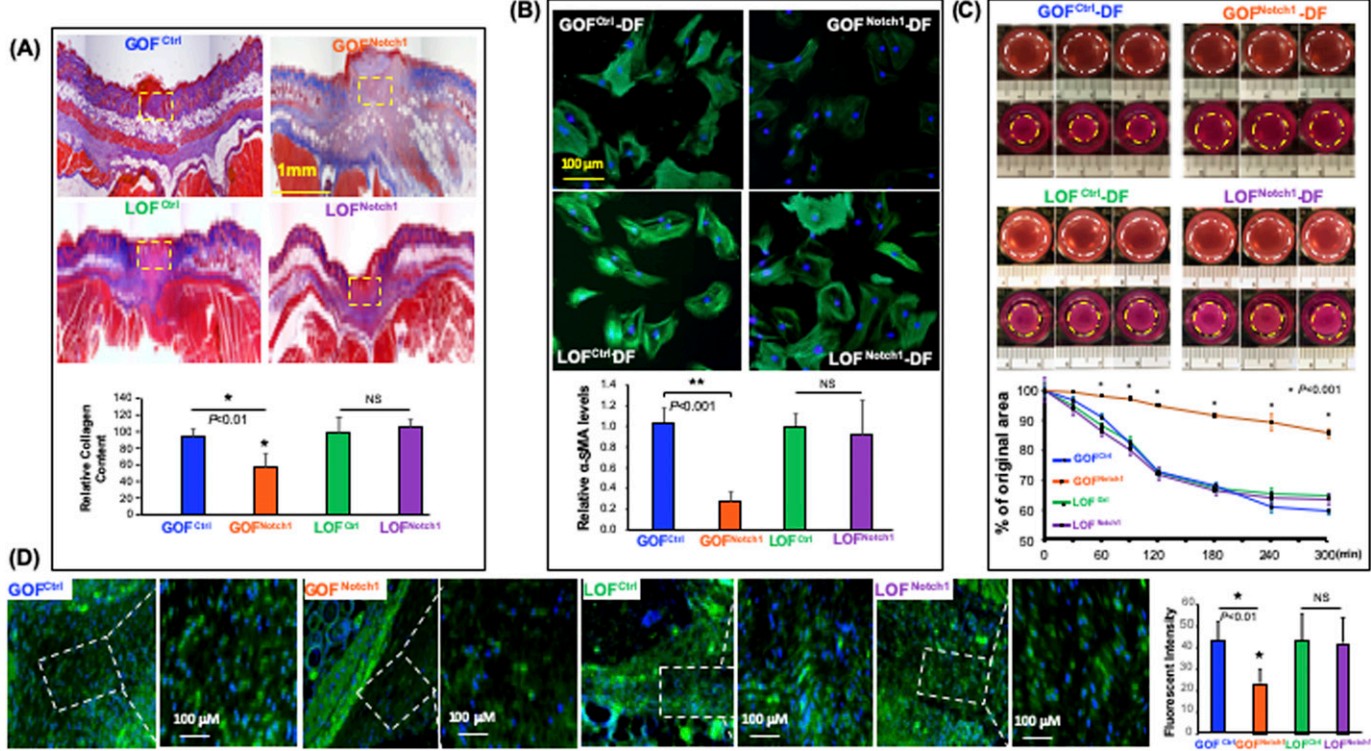

**Figure 4. Activation of the Notch1 pathway inhibits differentiation of fibroblasts into myofibroblasts.**
**(A)** Representative Masson's trichrome staining images show decreased collagen deposition in wound tissues of GOF[Notch1] mice compared to GOF[Ctrl] mice. No obvious difference in the wounds of LOF[Notch1] versus LOF[Ctrl] mice was found. Collagen levels in highlighted areas were quantified by ImageJ. Quantitative data of mean ± SD are based on results from three sections/wound (see N of wounds in Fig 3) and analyzed by $t$ test. **(B)** Immunostaining shows robustly decreased expression of $\alpha$-smooth muscle actin ($\alpha$-SMA) in GOF[Notch1]-(DF) dermal fibroblast compared with GOF[Ctrl]-DF, but no obvious difference in LOF[Notch1]-DF and LOF[Ctrl]-DF was found. Quantitative data are mean ± SD of intensity of green fluorescence of $\alpha$-SMA/cell based on total 100 cells in each group. **(C)** GOF[Notch1]-DF exhibited weak ability to contract the collagen gel. *Top*: representative three gel images/group of gel contraction assay in 0 and 5 h. *Bottom*: data are mean ± SD of sizes of six gels in each group compared to initial size (set as 100%) at 0 h. Experiments were repeated three times. **(D)** Immunostaining shows the levels of $\alpha$-SMA in myofibroblasts at wound granulation tissues of GOF[Notch1] versus GOF[Ctrl] and LOF[Notch1] versus LOF[Ctrl] mice. There were fewer myofibroblasts in a given area at wound granulation tissue of the GOF[Notch1] mice than GOF[Ctrl] mice. Also, (myo)fibroblasts at wound granulation tissue of GOF[Notch1] express lower amounts of $\alpha$-SMA than that in GOF[Ctrl] mice. No obvious difference between LOF[Notch1] and LOF[Ctrl] mice was found. Data are mean ± SD of numbers of myofibroblasts in selected given area with equal size and intensity of green fluorescence of $\alpha$-SMA/cell based on total 100 cells in each group (ANOVA). Intensity of green fluorescence is adjusted by blue signal intensity (DAPI signal) of each cell.

vascular branches were formed and grew into the center of the wounds in GOF[Ctrl] mice. Conversely, GOF[Notch1] contained immature vasculature (characterized by fewer, short and tortuous branches) that grew from edge of wounds towards the center of wounds. In addition, they contained leaky and hemorrhagic vessels (as evidenced by blurry vasculatures) in the center of wound beds. Immature and leaky vasculatures was reflected by ratio of DiI fluorescent signals in peripheral/center of wound beds. LOF[Notch1] versus LOF[Ctrl] mic showed no significant difference in wound angiogenesis (Fig 5B). These data indicate that activation of Notch1 pathway in fibroblasts results in decreased wound angiogenesis, suggesting a critical role of the Notch1 signaling in fibroblasts-modulated angiogenic response during wound healing.

Next, we tested the effect of the Notch1 activation in fibroblasts on modulating angiogenic response of ECs in a fibroblasts-modulated in vitro 3D angiogenesis model. This 3D model was developed to study vascular network formation by ECs under the support of fibroblasts embedded within the Type I collagen (Velazquez et al, 2002; Liu et al, 2003b). We embedded equal numbers of GOF[Notch1]-DF and GOF[Ctrl]-DF in collagen gels and compared side-by-side their

ability to support vascular network formation by human microvascular endothelial cells (HMEC). Compared with GOF[Ctrl]-DF, GOF[Notch1]-DF supported significantly less vascular network formation in 3D gels (Fig 5C). These results demonstrate that activation of the Notch1 pathway in fibroblasts diminished their capacity to modulate angiogenic response of ECs. Therefore, it indicates that decreased wound angiogenesis observed in GOF[Notch1] mice is ascribed to impaired function of DFs.

## Notch1 activation down-regulates IL-6 in fibroblasts

To explore the mechanism underlying Notch1-determined regulatory role of fibroblasts in angiogenesis, we investigated whether high intracellular Notch1 signaling in fibroblasts modulates the production of angiogenic factor(s). Thus, we conducted a ProteinArray analysis to assess the expression of a panel of angiogenic factors by human foreskin dermal fibroblasts (HDF) (Berking et al, 2001; Berking & Herlyn, 2001) in which the Notch1 pathway is constitutively activated by stable overexpression of N1[IC]-GFP using lentiviral vector and compared them with the control cells expressing GFP

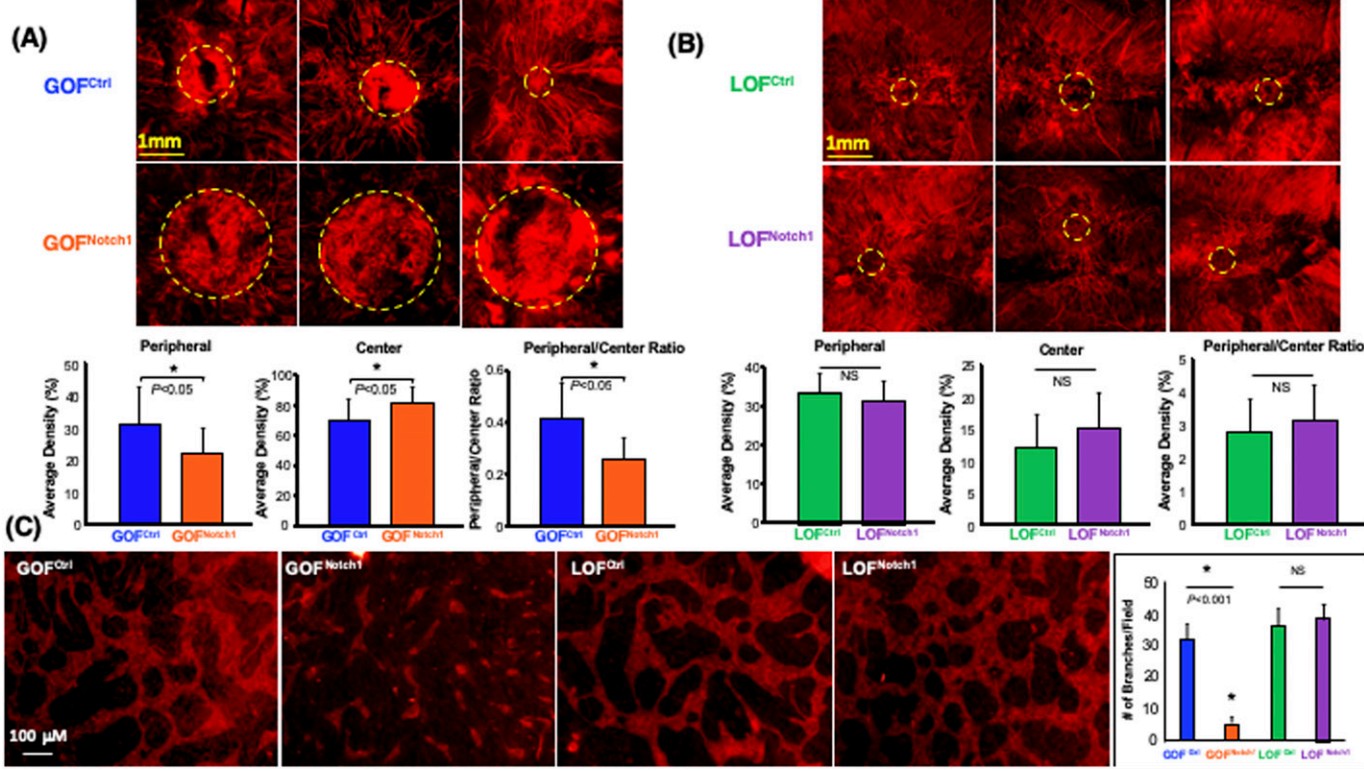

**Figure 5. Activation of the Notch1 pathway in fibroblasts inhibits the angiogenic response of endothelial cells.**
**(A)** Decreased neovascularization in the wounds of GOF[Notch1] mice compared with the wounds of GOF[Ctrl] mice. *Top*: representative three images of capillary networks developed in the wound beds. Centers of wound beds are highlighted by dash circles. *Bottom*: quantitative data are mean ± SD of intensity of red fluorescence signals of Dil in a given area in the center of wound bed in each group (n = 6/group). Ratio of peripheral/center reflects immature leaky vessels in the center of wound beds. **(B)** No significant difference in wound angiogenesis between LOF[Notch1] and LOF[Ctrl] mice was found. **(A)** The same displays in *Top* and *Bottom* as that in (A). **(C)** Inhibition of vascular network formation by GOF[Notch1]-DF. *Left*: representative images of capillary networks developed in 3D gel of fibroblasts-modulated in vitro angiogenesis assay. *Right*: quantitative data are mean ± SD of number of branches in a low power field (×0) of 3D gel (n = 6/group, ANOVA). Experiments were repeated three times.

using lentiviral vector (Fig 6A). HDF expressing GFP were sorted by FACS. The generated cells were named N1[IC]–GFP/HDF and GFP/HDF accordingly. The culture supernatants from N1[IC]–GFP/HDF and GFP/HDF were collected and subjected to TranSignal Angiogenesis Antibody Array (Panomics), which allows the detection of 48 different proteins. Levels of IL-6 were significantly down-regulated in N1[IC]–GFP/HDF compared with GFP/HDF (Fig 6B), suggesting that activation of Notch1 pathway in fibroblasts down-regulates production of IL-6. The decreased expression of IL-6 in cell lysates was confirmed by Quantikine Human IL-6 ELISA. The N1[IC]–GFP/HDF cell produced approximately a 2.5-fold lower amount of IL-6 compared with the GFP/HDF cell (Fig 6C). These data reveal that the Notch1 pathway activation down-regulates IL-6 production in human DFs.

To validate whether DFUF, in which Notch pathway is activated, express decreased levels of IL-6, we carried out ELISA to examine the levels of IL-6 in DFUF versus NFF. We confirmed that DFUF expressed lower levels of IL-6 than control fibroblasts (NFF) (Fig 6D). In addition, we conducted immunostaining to evaluate the amounts of IL-6 in wounds of GOF[Notch1] mice versus GOF[Ctrl] mice. We found that (myo)fibroblasts, which were stained with anti–FSP-1, in wound granulation tissues of GOF[Notch1] mice expressed less IL-6 compared with GOF[Ctrl] mice (Fig 6E). Together, our data

demonstrated that Notch1 pathway activation inhibits IL-6 production in DFs.

## IL-6 rescues decreased angiogenesis mediated by fibroblast Notch-1 activation in vivo and in vitro

To investigate whether down-regulation of IL-6 production in fibroblasts carrying high intracellular Notch activity accounts for their diminished role in modulating the angiogenic response of ECs, we tested the effect of exogenous IL-6 supplementation in a fibroblasts-modulated in vitro 3D angiogenesis model. Addition of recombinant human IL-6 (10 ng/ml) to collagen gel, in which N1[IC]–GFP/HDF were embedded, could partially rescue the vascular network formation (Fig 7A). These results indicated that decreased IL-6 is responsible for the inhibitory effect of Notch1-determined regulatory role of fibroblasts in modulating angiogenesis, suggesting that IL-6 is a functional down-stream target of Notch1 signaling in fibroblasts.

We further tested the effect of IL-6 in mediating fibroblasts-modulated angiogenesis using in vivo Matrigel plug assay. SCID mice were injected subcutaneously with 200 µl of growth factor reduced HC Matrigel Matrix in which $2 \times 10^5$ cells were embedded. Three groups were studied (N = 5/group): (i) GFP/HDF + Matrigel Matrix + PBS, (ii) N1[IC]–GFP/HDF + Matrigel Matrix + PBS, and (iii) N1[IC]–GFP/HDF + Matrigel Matrix + IL-6 (10 ng/ml).

Notch in fibroblasts and wound healing   *Shao et al.*   https://doi.org/10.26508/lsa.202000769   vol 3 | no 12 | e202000769   **7 of 16**

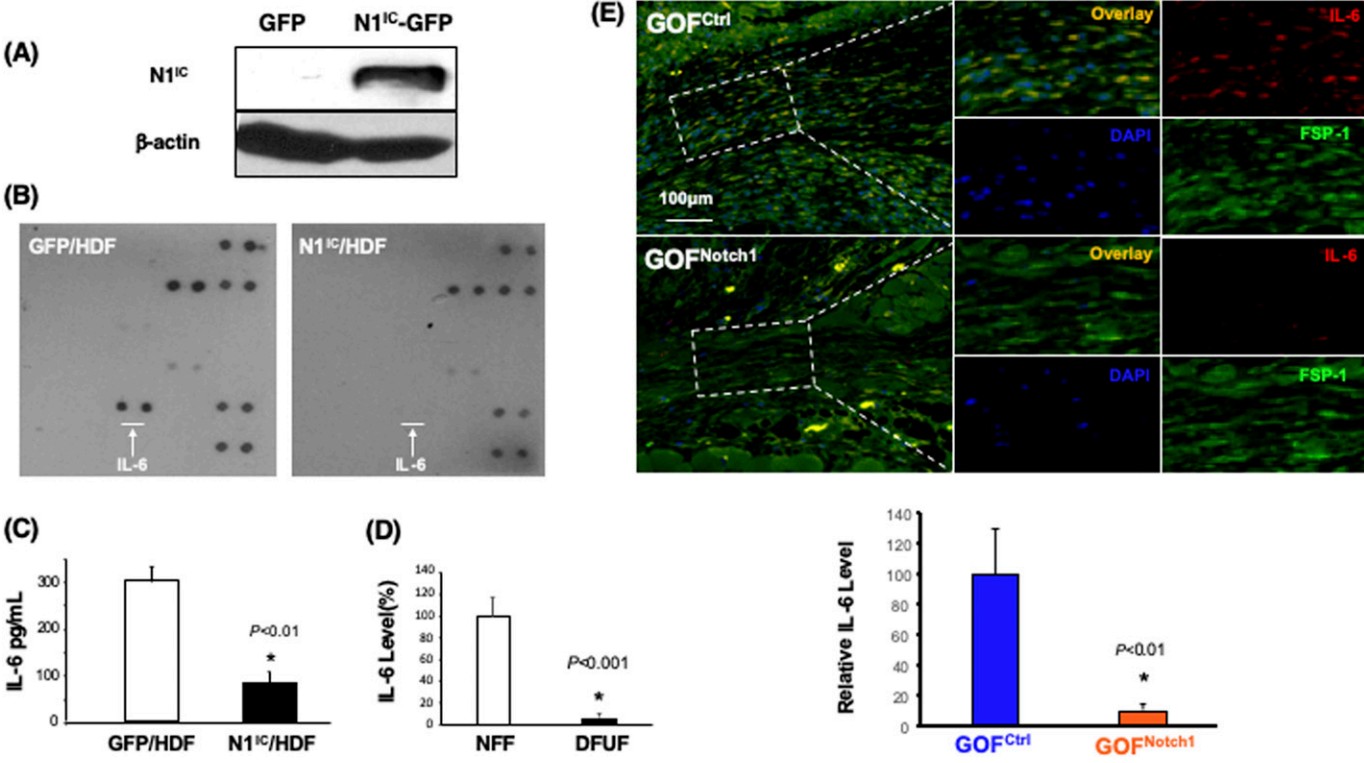

**Figure 6. Down-regulation of IL-6 production by activation of Notch1 pathway in fibroblasts.**
**(A)** Immunoblotting data show expression of N1$^{IC}$ protein in HDF transduced with N1$^{IC}$-GFP/Lentiviral and GFP/Lentiviral vectors, respectively. **(B)** Protein Array analysis displays that IL-6 production is down-regulated in HDF expressing N1$^{IC}$-GFP compared with HDF expressing GFP. White arrows point to the spots of IL-6 on the array membrane. **(C)** Quantitative data of ELISA analysis of IL-6 production (mean ± SD) of three independent experiments (*t* test). **(D)** ELISA shows decreased levels of IL-6 in cell lysates of three diabetic foot ulcer fibroblast compared with three NFF. Levels of IL-6 in NFF are set as 100%. Relative amount of IL-6 in diabetic foot ulcer fibroblast were calculated. **(E)** Immunostaining shows decreased levels of IL-6 (red) in myofibroblasts (FSP-1, green) at wound granulation tissues of GOF$^{Notch1}$ compared with the GOF$^{Ctrl}$ mice. *Top*: representative images of immunostaining. *Bottom*: Quantitative data of IL-6 production are calculated based on three sections/wound and mean ± SD are analyzed by *t* test.

10 d after injection, the Matrigel plugs were harvested and analyzed by immunohistochemistry. This assay has been used as a surrogate wound healing assay because the injected fibroblast-containing Matrigel plug is initially avascular (as is the early granulation tissue of a wound). Murine blood vessels growing into the Matrigel plug were stained with antimouse CD31 antibody. The vessel density in explanted plugs was determined by counting the number of blood vessels in five randomly selected fields from each Matrigel plug. Consistent with observed results from in vitro 3D angiogenesis assays, N1$^{IC}$–GFP/HDF significantly inhibited neovascularization into the Matrigel plug, whereas supplemental IL-6 was able to reverse the inhibitory effect of N1$^{IC}$–GFP/HDF on neovascularization (Fig 7B). These in vivo results confirm that activation of the Notch1 pathway in fibroblasts attenuates their function in supporting angiogenesis and indicates that Notch1-determined fibroblasts' regulatory effect on angiogenesis is mediated, at least in part, by down-regulation of IL-6 expression.

## Discussion

Diabetic, non-healing wounds are a major clinical problem with considerable morbidity and associated financial costs. However, mechanisms by which diabetes impedes tissue repair mechanisms remain unclear. Previous studies have suggested decreased tissue levels of growth factors, including keratinocyte growth factor, VEGF, PDGF, excess protease activity, decreased angiogenesis, altered inflammation, or an increased microbial load as possible contributing factors for the impaired wound healing observed in diabetes mellitus (Galkowska et al, 2006; Brem & Tomic-Canic, 2007; Grice et al, 2010; Gardner et al, 2013; Eming et al, 2014; Pastar et al, 2014; Lindley et al, 2016; Quinn et al, 2016; Ramirez et al, 2018). In this study, we discovered that the Notch pathway activity is elevated in fibroblasts of human diabetic ulcers and diabetic murine wounds, but not in normal murine acute wounds and non-diabetic ischemic wounds. Furthermore, we uncovered the Notch1 pathway as an important molecular determinant in controlling the plasticity and function of fibroblasts' role in modulation of diabetic wound healing and angiogenesis. We demonstrate that a dysregulated intracellular Notch1 pathway is responsible for the impaired plasticity of fibroblasts and fibroblasts-modulated wound healing and angiogenesis in various in vitro and in vivo models. The intracellular Notch1 signaling pathway in fibroblasts may, therefore, serve as a potential target for therapeutic interventions in diabetic wound healing.

The formation of granulation tissue, which is comprised of new connective tissue rich in myofibroblasts and newly formed microscopic

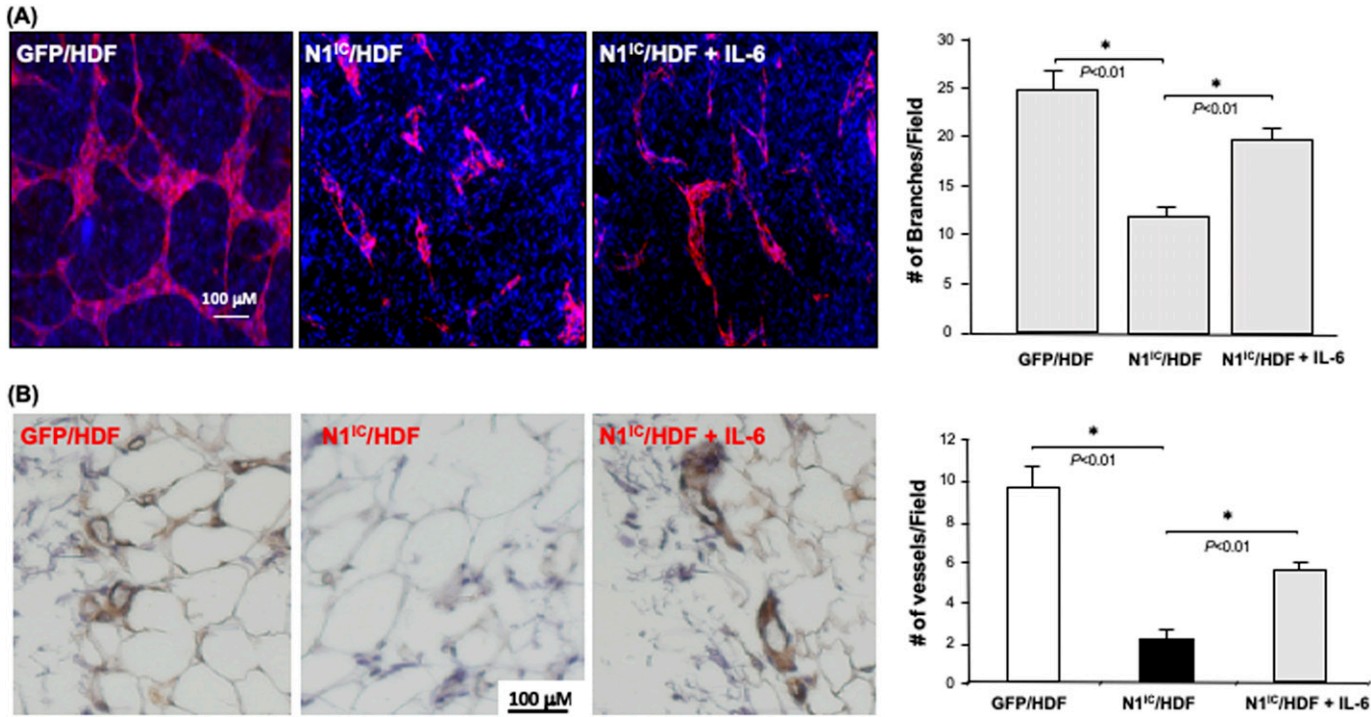

**Figure 7.  Rescue of capillary network formation with supplemental IL-6.**
**(A)** *Left*: representative images of capillary networks in in vitro 3D angiogenesis assay. Exogenous supplemental γhIL-6 can partially rescue the vascular network formation modulated by N1[IC]–GFP/HDF. *Right*: quantitative data of vascular network formation by endothelial cells in 3D angiogenesis assay. Data are analyzed by one-way ANOVA followed by post-hoc test and presented as mean ± SD of three independently performed experiments. **(B)** Supplemental γhIL-6 in Matrigel reverses the inhibitory effects of N1[IC]–GFP/HDF on angiogenesis in mouse Matrigel plug model. *Left*: representative images of IHC. Blood vessels are brown color (DAB) *Right*: quantitative data of vessel density. Data are analyzed by one-way ANOVA followed by post-hoc test and presented as mean ± SD of five randomly selected fields from each mouse/Matrigel plug in a given group (5 mice/group) (×40).

blood vessels, is essential for cutaneous wound healing. The myofibroblasts are central to this healing process. They function as both builders which deposit a collagen-rich matrix and orchestrators that coordinate tissue repair and regeneration by secreting numerous cytokines and growth factors important for cell–cell communication (Ryan et al, 1974; Takehara, 2000; Tracy et al, 2016). Any impediment to the quantity or quality of myofibroblasts will interfere with normal wound healing and may result in a chronic non-healing wound. The differentiation of fibroblasts to myofibroblasts is one of key events in wound healing. When tissues encounter traumatic events, fibroblasts will undergo a phenotypic change from their default, relatively quiescent state (in which they are involved in the slow turnover of the ECM) to a proliferative and contractile phenotype as a myofibroblast. α-SMA is a well-recognized marker of myofibroblasts. Other features of myofibroblasts include the production of several components of the ECM, such as collagens (Schurch et al, 1981; Seemayer et al, 1981) and fibronectin and are highly contractile (Torr et al, 2015). Our findings demonstrate that intracellular Notch1 pathway activity regulates the differentiation of fibroblasts into myofibroblasts as we observed that high intracellular Notch1 pathway activity in fibroblasts results in the suppression of α-SMA expression, decreased collagen production and low contractility. These findings not only unveil one of the molecular mechanisms of

impaired diabetic wound healing, but also reveals new role of Notch1 signaling in regulating cell fate.

The Notch pathway is an evolutionarily conserved signaling cascade that regulates a variety of cellular activities including proliferation, differentiation, quiescence and cell death (Yin et al, 2010; Shao et al, 2012). The role of Notch signaling in fibroblasts was not well delineated previously. Studies from our laboratory and others imply that Notch signaling serves as a negative regulator or "break" on the growth of fibroblasts. We observed that loss of *Notch1* in MEFs promotes cell growth and migration, whereas Notch1 activation inhibits cell growth and motility of human fibroblasts (Liu et al, 2012). Other investigators have reported that the Notch pathway activation resulted in cell-cycle arrest and apoptosis in MEFs (Ishikawa et al, 2008). These previous studies suggest that high intracellular Notch pathway activity attenuates cellular activity of fibroblasts and are consistent with the findings of the current study. Interestingly, the inhibitory role of Notch pathway activation in modulating keratinocyte-, EC-, and macrophages-mediated diabetic wound healing has been reported (Kimball et al, 2017; Zheng et al, 2019). These findings, along with ours, imply a general role of the Notch signaling in the regulation of the reparative responses of various types of cells in diabetic wound healing.

On the other hand, tumors have been described as wounds that do not heal (Dvorak, 1986). Tumors are highly complex tissues composed

of cancer cells and stromal cells, including fibroblasts termed tumor stromal fibroblasts or cancer-associated fibroblasts (CAFs) (Orimo and Weinberg, 2006, 2007; Kalluri, 2016). Like wound healing, tumor progression and metastasis are also tightly regulated by CAFs (Orimo & Weinberg, 2006; Kalluri, 2016; LeBleu & Kalluri, 2018). In various tumor models, we have consistently observed that the Notch1 signaling pathway functions as a crucial molecular determinant in governing fibroblasts' regulatory role in tumor progression and metastasis. Elevated Notch1 pathway activity inhibits the function of CAFs in promoting tumor progression. For example, our prior work demonstrated that co-grafted normal skin fibroblasts, which were pre-engineered to carry high Notch1 activity, inhibited tumor growth and angiogenesis in a tumor xenograft model (Shao et al, 2011), revealing that Notch activation antagonizes the tumor-promoting effect of stromal fibroblasts. We also showed that CAFs carrying elevated Notch1 activity significantly inhibited tumor growth and invasion, whereas those with a null Notch1 activity promoted tumor invasion (Shao et al, 2015). Hence, these previous results generated from tumor models are also consistent with data derived from wound healing models in terms of the role of Notch1 pathway in determining function and cellular activity of fibroblasts.

Interestingly, inactivation of the Notch pathway by deletion of Notch1 in fibroblasts has little effect on cellular behaviors and fibroblast-modulated wound healing response. The mechanisms underlining unaltered wound healing rates between LOF^Notch1 versus LOF^Ctrl mice and cellular behavior between LOF^Notch1 versus LOF^Ctrl remain unclear. It is possible that the Notch1 signaling is maintained in an activated status (the Notch1 signaling is "OFF") in skin fibroblasts in normal mice as evidenced in Fig 1D (C57B6), Figs S4–S6, and LOF^Ctrl mice as evidenced in Fig S7, deletion of *Notch1* will not bring about additional effect.

Active formation of new blood vessels is another characteristic of healthy granulation tissue. Neovascularization plays a critical role in wound healing (Folkman, 1995). Angiogenesis is a dynamic cellular response that requires temporal and spatial regulation of multiple cell types (Bauer et al, 2005; Velazquez, 2007). The assembly and stabilization of vascular networks are largely dictated by external signals from surrounding stromal cells (e.g., fibroblasts) within the local microenvironment (Velazquez et al, 2002). Increasing evidence suggests that fibroblasts are an important component of stroma-modulated angiogenesis and provide a unique microenvironment that contributes to the organization and maintenance of the elaborate post-natal microvasculature (Orimo & Weinberg, 2006; Hughes, 2008). Our finding of IL-6 as a functional downstream mediator of Notch1 signaling in regulating wound angiogenesis reveals a paracrine mechanism by which fibroblasts communicate with ECs and modulate angiogenesis. IL-6 is a potent proinflammatory cytokine and participates in angiogenesis during wound healing, tumor progression, and the development of the cerebral vasculature (Fee et al, 2000). IL-6 has been known to activate signal transducers and activators of transcription 3 (STAT-3) signal pathway by binding to the gp130 subunit, which then transduces intracellular signals and produces various biologic functions (Kishimoto et al, 1995). This signal pathway widely exists in human ECs and mediates several pathological post-natal neovascularization processes (Seino et al, 1994). Overexpression of IL-6 in the central nervous system is correlated to pronounced vascularization in vivo

(Campbell et al, 1993). Increased IL-6 is observed in patients with giant-cell arthritis, indicating that IL-6 activates a functional program related to pro-inflammatory angiogenesis (Hernandez-Rodriguez et al, 2003). IL-6 is also increased in patients after a cerebral vascular accident, which may reflect these patients' change in inflammatory-angiogenesis status (Salobir & Sabovic, 2004; Lobbes et al, 2006). IL-6 promotes angiogenesis via MMP-9 activation which induces release of VEGF from cultured ECs and tumor cells (Cohen et al, 1996; Yao et al, 2006) and also induces expression of VEGF-R2 (KDR) on cultured ECs (Cohen et al, 1996). Moreover, IL-6 induces expression of decorin, a small multifunctional proteoglycan expressed by sprouting ECs during inflammation-induced angiogenesis in vivo and by human ECs co-cultured with fibroblasts in a collagen lattice (Strazynski et al, 2004). Decreased IL-6 production by fibroblasts carrying high Notch1 pathway activity explain, at least in part, the down-regulated angiogenesis observed in both human and murine samples of our study. Therefore, our study highlights the intracellular Notch1 pathway in fibroblasts to be a potential target for therapeutic intervention in diabetic wound healing. However, it remains unknown whether Notch1 signaling regulates IL-6 through direct or indirect mechanisms and what other key factors are contributing to the observed inhibition of angiogenesis by fibroblasts carrying high Notch1 pathway activity.

In contrast to understanding the downstream targets of the Notch1 signaling pathway, the upstream mechanisms for the Notch pathway activation in diabetic fibroblasts remain unknown. Activation of the Notch pathway is a dynamic process in fibroblasts during diabetic wound healing. The activated intracellular Notch signaling in fibroblasts fades away and switches "OFF" when the diabetic wounds are healed. Dynamic changes in fibroblasts' intracellular Notch pathway activity during the diabetic wound healing process suggest a crucial influence from wound tissue microenvironments in diabetes mellitus. The question of how pathological conditions presented in DFU tissues, such as hyperglycemia, hypoxia, oxidative stress and inflammation increase intracellular Notch pathway activity in fibroblasts has been raised and will be studied in the future.

Fibroblasts also play pivotal roles in tissue remodeling. In the later remodeling stage of wound healing, myofibroblasts undergo apoptosis resulting in a decreased cellular density to avoid fibrosis and excessive matrix deposition that result in the overgrowth of tissue, hardening, and scar formation. Hence, development of any therapies for the treatment of wound healing through correction and improvement of the function of skin fibroblasts needs to avoid fibrosis and scarring. Modulation of Notch-1 activity seems to have all attributes of such therapeutic approach. Induced activity of intracellular Notch1 inhibits wound healing and angiogenesis, but eliminating Notch1 signaling does not (as shown in LOF^Notch1 mice). Furthermore, no evidence of fibrosis, scarring or excessive matrix production was found in LOF^Notch1 mice or LOF^Ctrl mice. Thus, our preclinical in vivo data suggest that targeting Notch1 activity in fibroblasts to modulate its excessive activity appears to be safe and may have significant therapeutic potential for patients with non-healing DFUs.

In conclusion, identified connections between the intracellular Notch pathway activity and the plasticity and biologic activity of

fibroblasts carry significant clinical relevance. Potentially, the elevated Notch pathway activity in fibroblasts within the micro-environment of a chronic diabetic wounds could be manipulated and reduced to achieve the desired positive effects on fibroblast proliferation, migration, differentiation into myofibroblasts, and angiogenic support. Future studies are warranted to test and verify the precise targeted modulation of the Notch pathway activity in fibroblasts of chronic diabetic wound tissues as a novel therapeutic approach in the treatment of diabetic non-healing ulcers.

# Materials and Methods

### Reagents

Type I collagen was purchased from Organogenesis; Recombinant human IL-6 protein was purchased from R&D Systems. SDS–polyacrylamide gels were obtained from Invitrogen. All other chemicals and solutions were from Sigma-Aldrich unless otherwise indicated.

### Mice

$Notch1^{Loxp/LoxP}$ mice were described (Radtke et al, 1999). $ROSA^{LSL-N1IC+/+}$ (#008159) and $Fsp1.Cre^{+/-}$ (#012641) mice were purchased from The Jackson Lab. All these mice have a C57 BL6 background. The Gain-Of-Function Notch1 ($GOF^{Notch1}$: $Fsp1.Cre^{+/-};ROSA^{LSL-N1IC+/+}$) and Loss-Of-Function Notch1 ($LOF^{Notch1}$: $Fsp1.Cre^{+/-};Notch1^{LoxP/LoxP+/+}$) lines were generated by crossing $ROSA^{LSL-N1IC+/+}$ and $Notch1^{Loxp/LoxP+/+}$ with $Fsp1.Cre^{+/-}$ mice, and subsequently crossing $Fsp1.Cre^{+/-};ROSA^{LSL-N1IC+/-}$ with $ROSA^{LSL-N1IC+/+}$ mice and $Fsp1.Cre^{+/-};Notch1^{LoxP/LoxP+/-}$ with $Notch1^{Loxp/LoxP+/+}$ mice, respectively. $GOF^{Ctrl}$ ($FSP1.Cre^{-/-};ROSA^{LSL-N1IC+/+}$) and $LOF^{Ctrl}$ ($FSP1.Cre^{-/-}; Notch1^{LoxP/LoxP+/+}$) mice were used as control. C57 BL6 (#000664), NOD (#001976) and db/db (#000697) mice were also purchased from The Jackson Lab. Mice were maintained at the Division of Veterinary Resources (DVR) animal facility under standard conditions. Mice were anesthetized for all surgical procedures by ketamine/xylazine mixture (100/10 mg/kg, IP), and imaging procedures by inhaling 3% isoflurane gas, and euthanized in $CO_2$ chamber.

### Human and mouse skin and wound tissues

Full-thickness skin and DFU samples obtained from consenting donors and patients receiving standard care at the University of Miami Hospital. See Table S1 for basic clinical information of the human subjects. Parts of fresh human tissues were used for isolation of primary cells and remaining tissues were fixed in 10% formalin (Sigma-Aldrich), and 5-μm paraffin sections were used for immunostaining (three samples used for the isolation of cell lines were that for immunostaining). Dorsal skin wound were created by 6-mm punch biopsy on 4-wk-old $GOF^{Notch1}$, $GOF^{Ctrl}$, $LOF^{Notch1}$, and $LOF^{Ctrl}$ mice. Male and female mice are 50%:50% and randomly selected. Mouse dorsal skin wound (created by 6-mm punch biopsy) tissues were obtained from 10- to 15-wk-old C57 BL6 (non-ischemic), 28- to 33-wk-old diabetic NOD (type I diabetes), 18- to 20-wk-old diabetic db/db (type II diabetes) mice, and ischemic skin wound (4-mm skin punch biopsy on the anterior thigh of ischemic limb created by femoral artery ligation [Parikh et al, 2018]) from 10- to 15-wk-old C57 BL6 mice. Blood glucose levels of NOD and db/db mice >250 mg/dl for three consecutive days were considered diabetes. Mean serum glucose levels in diabetic mice were 423 mg/dl with a range of 326–527 mg/dl, whereas mean serum glucose levels in C57 BL6 mice were 115 mg/dl with a range of 85–148 mg/dl. Serum glucose was measured from the mouse tail vein using a glucometer. NOD and db/db mice developed diabetes at 16~24 wk old. All C57 BL6, db/db, and NOD mice are female.

### Cells and cell culture

Primary fibroblasts were generated from discarded foot skin specimens collected from routine procedures, such as bunionectomy, phalangectomy, or arthroplasty according to (IRB) protocols #20140473 and B# 20120574. Cells were representative of two groups of donors: DFUFs from three diabetic individuals with a non-healing foot ulcer at the site of wound edge and non-diabetic normal foot fibroblasts (NFF) from three healthy, non-diabetic donors. See Table S1 for basic clinical information of the human subjects. Patient demographics and isolation of primary fibroblasts from skin specimens was also previously described (Ramirez et al, 2015, 2018; Liang et al, 2016; Maione et al, 2016; van Asten et al, 2016). Briefly, skin samples were treated in dispase (Roche) overnight at 4°C and then centrifuged to collect any released cells followed by removal of the epidermis (keratinocytes) from the dermis the following day based on established protocol (Normand & Karasek, 1995). Subsequently, the dermis was cut into small pieces, and then treated with collagenase and hyalurondiase in DMEM-F12 (Invitrogen) for 1 h at 37°C with stirring. The cell suspension was mixed with red blood cell lysis buffer, centrifuged, and then cells were collected and plated. Fibroblasts were grown in 1 g/l glucose DMEM (Invitrogen), 10% FBS (HyClone), Hepes (Sigma-Aldrich), and Pen/Strep/Fung (Invitrogen), passaged after reaching confluence, and second passage stocks were frozen in liquid nitrogen. Selective growth conditions were also used to remove macrophages or any resident immune cells, and the purity of fibroblasts was confirmed by flow cytometry analysis based on positive staining for vimentin and CD-140α (PDGFR), and negative staining for CD31 and CD45. Protein extraction was performed at passages 1–3. Human foreskin dermal fibroblasts were described previously (Berking et al, 2001; Berking & Herlyn, 2001). Fibroblasts were maintained in low-glucose DMEM containing in 1 g/l glucose (Invitrogen), 10% fetal bovine serum (HyClone), Hepes (Sigma-Aldrich), and Pen/Strep/Fung (Invitrogen). Human microvascular endothelial cells (HMVEC) were purchased from ATCC (CRL-4025) and cultured in complete M199 medium (Invitrogen). 293T and NIH/3T3 cells were also cultured in complete DMEM.

### Recombinant lentiviruses and viral infection of targeting cells

Generation of GFP/lenti and N1$^{IC}$–GFP/lenti using 293T cells were described previously (Balint et al, 2005). Lentiviruses collected 48 h post-transfection displayed titers of around $10^7$ transducing units/ml in NIH/3T3 cells. To infect HDF by lentiviruses, cells were exposed

to virus with MOI five in the presence of 4 $\mu$g/ml polybrene for 6 h. Cells were then washed and cultured with regular complete medium for two additional days. GFP$^+$ cells were then sorted by FACS (FACSAria II; BD Biosciences).

## Cell proliferation assay, in vitro wound healing assay, and collagen gel contraction assays

Cell proliferation was tested using the WST cell proliferation kit (BioVision) according to the manufacturer's instruction. $5 \times 10^3$ cells were plated on 96-well plates and cultured for overnight before the WST assay. In vitro wound healing assay was performed using radius 24-well cell migration assay kit (CBA-125; Cell Biolabs, Inc.) according to the manufacturer's instruction. Collagen gel contraction assay was conducted using cell contraction assay kit (CBA-201; Cell Biolabs, Inc.) according to the manufacturer's instruction. All assays were tested in triplicates and assays were repeated three times.

## Mouse skin wound healing model

Skin wounds were induced on the dorsal surface of the mouse back using a 6-mm punch biopsy. Full-thickness skin was removed, exposing the underlying muscle. Wounds were followed serially with daily digital photographs using an Olympus digital camera. A ruler was included in all photos to allow for calibration of measurements. Images were analyzed using ImageJ software (Imaging Processing and Analysis in Java, National Institutes of Health). Wound area was measured each day, and the wound's percent recovery rate was expressed as [(original wound area minus daily wound area)/(original wound area)] × 100.

## Live animal blood vessel perfusion and laser scanning confocal microscopy

Mouse blood vessels were directly labeled in vivo in anesthetized mice by live perfusion using a specially formulated aqueous solution (7 ml/mouse) containing DiI (D-282; Invitrogen/Molecular Probes), which incorporates into EC membranes upon contact, and was administered via direct intra-cardiac injection before animal euthanasia as previously reported (Li et al, 2008; Shao et al, 2011). 7 ml of fixative (4% paraformaldehyde) was injected after DiI perfusion, and the entire wound tissue was harvested. The vascular network was visualized by scanning the entire wound tissue to a thickness or depth of 200 $\mu$m, using laser scanning confocal microscopy (Vibratome [VT1000S; Leica Microsystems]). Vessel density was quantified assessing total number of red DiI-labeled vessels normalized to the entire scanned wound area, using ImageJ software.

## Immunoblotting, immunohistochemistry, Masson's trichrome staining, antibody array, and ELISA

Immunoblotting was performed as described (Shao et al, 2011, 2016). Membranes were probed with Abs to activated Notch1 (ab8925), Notch1 (ab52627), Notch2 (ab72803), Notch3 (ab23426), Dll1 (ab84260), Hes1 (ab71559), Hey1 (ab22614) from Abcam, Notch4

(OASG05123; Aviva Systems Biology), Jag1 (AP091279U-N), Jag2 (AP13110PU-N), Dll3 (AP21739PU-N) from OriGene Technologies, Inc., Dll4 (PA5-97664; Thermo Fisher Scientific), and IL-6 (NB600-1131; Novus). For immunohistochemistry, 5-$\mu$m paraffin sections were processed as described (Shao et al, 2011, 2016) and incubated with anti-Hes1 (ab71559), Ki67 (ab15580), $\alpha$-SMA (ab7817) from Abcam, and FSP-1 (NBP2-52890; Novus) and corresponding isotype matched non-immunogen antibodies overnight at cold room, and then incubated with secondary antibodies from Invitrogen (Goat anti-Mouse IgG Alexa Fluor 488 conjugated for FSP-1 and $\alpha$-SMA, Donkey anti-Rabbit IgG Alexa Fluor 594 conjugated for Hes-1 and Ki67). The nuclei were stained with DAPI (Vector Laboratories). Masson's trichrome staining was performed as described previously (Gallagher et al, 2007). The relative levels of $\alpha$-SMA/Hes-1/Hey-1/N1$^{IC}$ expression were determined using ImageJ software. The fluorescence-integrated intensity of Hes-1/Hey-1/N1$^{IC}$ in each FSP-1–positive cells was measured and divided by its own FSP-1 integrated intensity. For $\alpha$-SMA, its integrated intensity was divided by its nuclear DAPI intensity. Then each cell's relative levels of $\alpha$-SMA/Hes-1/Hey-1/N1$^{IC}$ expression were grouped, and average of relative fluorescence intensity was calculated.

To do antibody array, cell lysates were diluted with PBS to adjust concentration to 10 $\mu$g/$\mu$l. Equal amount of proteins (200 $\mu$g) were subjected to array analysis with TranSignal Angiogenesis Antibody Array from Panomics, which allows detection of 48 different proteins, based on the manufacturer's protocol. Concentration of IL-6 was measured by Quantikine IL-6 ELISA kit (R&D Systems) based on the manufacturer's protocol. Briefly, $3 \times 10^4$ cells/well were plated and grew in 96 well plates for overnight and levels of IL-6 in the supernatant of cell culture or cell lysates were tested by ELISA kit.

To inhibit Notch pathway activation, DFUFs were treated with DAPT at indicated concentrations (DAPT [#D5942; Sigma-Aldrich]) and 5 $\mu$g/ml Jagged-1–neutralizing Ab (PA5-46970; Thermo Fisher Scientific) for 48 h, respectively. Cells were subjected to immunoblotting analysis.

## In vitro 3D angiogenesis assay

Formation of vessel-like structures in 3D collagen gels and subsequent fluorescent staining of networks/cords in whole-mounted gels were performed as previously described (Velazquez et al, 2002; Liu et al, 2003a). Briefly, HMVEC were cultured as monolayers on bovine type I collagen-coated 24-well plates at $1 \times 10^5$ cells/well for 24 h and overlaid with acellular collagen mixed in 10 × medium 199 with heparin (100 U/ml), vitamin C (50 $\mu$g/ml), and FBS (1%). After polymerization of the collagen gels, cells were further overlaid with a second collagen layer containing $5 \times 10^5$ cells/ml N$^{IC}$–GFP/HDF versus GFP/HDF cells. Wells were then filled with MCDB131 containing 5% FBS. The reconstructs were incubated at 37°C for 5 d. To prepare for staining, medium was removed, and the collagen gels were fixed in Prefer (Anatech LTD) for 4 h at room temperature. Gels were processed as whole-mounts. After blocking with 10% goat serum, gels were stained with monoclonal anti-vWF VIII Ab followed by a PE-conjugated second Ab (Jackson Immunoresearch). Staining of EC networks/cords was examined by inverted fluorescence microscopy and gels were photographed. The nuclei were counterstained with DAPI.

### In vivo Matrigel plug assay

8-wk-old female SCID CB-17 mice were purchased from Charles River Laboratories. Animal experiments were approved by the Institutional Animal Care and Use Committee (IACUC) of University of Pennsylvania (protocol# 801110) which conforms to the Guide for the Care and Use of Laboratory Animals published by the U.S. National Institutes of Health (NIH Publication No. 85-23, revised 1996). Mice were injected subcutaneously with 200 $\mu l$ of mixture of growth factor reduced BD Matrigel Matrix High Concentration (BD Biosciences) with $2 \times 10^5$ N$^{IC}$–GFP/HDF with or without IL-6 (10 ng/ml) and GFP/HDF cells, respectively. 10 d after injection, mice were euthanized in $CO_2$ gas chamber and Matrigel plugs were harvested, fixed in 10% formalin/PBS, embedded in paraffin and sectioned. 5-$\mu m$ sections were subjected to immunohistochemistry analysis.

### Statistics

Statistical analysis of differences was performed using ANOVA followed by post-hoc test (for multiple groups comparison) and two-tail Student's $t$ test (for paired comparison). Data were analyzed using Microsoft Excel (Microsoft Corp). Data are expressed as mean ± SE. Values are considered statistically significant when $P <$ 0.05.

### Study approval

All animal experimental procedures were carried out with approval from the University of Miami Institutional Animal Care and Use Committee (IACUC, protocol# 17-130), except in vivo Matrigel plug assay, which was approved by the IACUC of University of Pennsylvania (protocol# 801110). Human skin and DFU samples obtained from donors and patients receiving standard care at the University of Miami Hospital. The written informed consent was received from participants before inclusion in the study. The protocols, including written informed consent, were approved by the University Institutional Review Board (IRB) (protocols IRB #20140473 and IRB# 20120574).

### Supplementary material

All materials are described in "the Materials and Methods section."

# Data Availability

The data that support the findings of this study are available on request from the corresponding author (Z-J Liu).

# Supplementary Information

# Acknowledgements

We thank Dr Hallie J Quiroz for the critical editing of this manuscript. We also thank Dr F Radtke (Swiss Institute for Experimental Cancer Research) for providing us Notch1$^{Loxp/LoxP}$ mice and Dr Alice A Tomei (Department of Biomedical Engineering, University of Miami) for providing some NOD mice. This work was supported by grants from the National Institutes of Health (R01DK-071084, R01GM081570 to OC Velazquez, R01HL149452 to Z-J Liu and OC Velazquez, and R01NR015649, R01NR13881 to M Tomic-Canic).

## Author Contributions

H Shao: data curation, formal analysis, validation, investigation, methodology, and writing—review and editing.
Y Li: data curation, formal analysis, validation, investigation, and methodology.
I Pastar: resources, data curation, formal analysis, validation, investigation, visualization, methodology, and writing—review and editing.
M Xiao: data curation, formal analysis, investigation, and methodology.
R Prokupets: data curation, formal analysis, investigation, and methodology.
S Liu: data curation, investigation, and methodology.
K Yu: data curation, investigation, and methodology.
RI Vazquez-Padron: conceptualization, formal analysis, and writing—review and editing.
M Tomic-Canic: conceptualization, supervision, funding acquisition, project administration, and writing—review and editing.
OC Velazquez: conceptualization, resources, supervision, funding acquisition, project administration, and writing—review and editing.
Z-J Liu: conceptualization, resources, supervision, funding acquisition, validation, investigation, project administration, and writing—original draft, review, and editing.

## Conflict of Interest Statement

The authors declare that they have no conflict of interest.

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
