## [Reviewer comments · Life Science Alliance]

Life Science Alliance

Notch1 Signaling Determines the Plasticity and Function of Fibroblasts in Diabetic Wounds

Hongwei Shao, Yan Li, Irena Pastar, Min Xiao, Rochelle Prokupets, Sophia Liu, Kerstin Yu, Roberto Vazquez-Padron, Marjana Tomic-Canic, Omaid Velazquez, and Zhao-Jun Liu

DOI: <https://doi.org/10.26508/lsa.202000769>

Corresponding author(s): Zhao-Jun Liu, University of Miami

Review Timeline:

Submission Date:	2020-05-07
Editorial Decision:	2020-06-16
Revision Received:	2020-09-25
Editorial Decision:	2020-10-14
Revision Received:	2020-10-16
Accepted:	2020-10-16

Scientific Editor: Shachi Bhatt

Transaction Report:

June 16, 2020

Re: Life Science Alliance manuscript #LSA-2020-00769-T

Dr. Zhao-Jun Liu
University of Miami
Molecular and Cellular Pharmacology
1600 NW 10th Ave
RMSB Bldg, RM1048
Miami, Florida 33136

Dear Dr. Liu,

Thank you for submitting your manuscript entitled "Notch1 Signaling Determines Fibroblast Differentiation and Fibroblasts-Modulated Angiogenic Response in Diabetic Wounds" to Life Science Alliance. The manuscript was assessed by expert reviewers, whose comments are appended to this letter.

As you will see, while the reviewers appreciate the interesting findings presented in your manuscript, they also raise numerous concerns. We do not rule out however that you might be able to address a majority of these points, and given the overall interest in your study, we would still like to invite you to submit a revised version. We would be happy to discuss the individual revision points further with you should this be helpful. When submitting the revision, please include a letter addressing the reviewers' comments point by point.

In our view these revisions should typically be achievable in around 3 months. However, we are aware that many laboratories cannot function fully during the current COVID-19/SARS-CoV-2 pandemic and therefore encourage you to take the time necessary to revise the manuscript to the extent requested above. We will extend our 'scooping protection policy' to the full revision period required. If you do see another paper with related content published elsewhere, nonetheless contact me immediately so that we can discuss the best way to proceed.

Please note that papers are generally considered through only one revision cycle, so strong support from the referees on the revised version is needed for acceptance.

Thank you for this interesting contribution to Life Science Alliance. We are looking forward to

receiving your revised manuscript.

Sincerely,

Reilly Lorenz
Editorial Office Life Science Alliance
Meyerhofstr. 1
69117 Heidelberg, Germany
t +49 6221 8891 414
e contact@life-science-alliance.org
www.life-science-alliance.org

B. MANUSCRIPT ORGANIZATION AND FORMATTING:

Reviewer #1 (Comments to the Authors (Required)):

In this paper the authors examine the role of Notch 1 in fibroblasts on angiogenesis, particularly in

the setting of diabetic wounds. They identify that IL-6 is a functional Notch1 target and involved in regulating angiogenesis. The paper is well written and the experiments justify the conclusions- several novel mouse models were used as well as human diabetic wound tissues to increase the translational potential of these findings. The data are timely and very important in the field of diabetic wound healing. I have the following comments:

1. For Figure 1C why was Day 7 chosen? Is there a change in Notch1 signaling any earlier? Does it persist in the diabetic setting? It would be interesting to see various time points and whether there is a sustained increase in the diabetic setting compared to control.
2. IN the Notch 1 GOF mice, does notch 1 activation in fibroblasts have any effect on the other cells (both structural and immune) - in other words, does the cellular milieu change in response to notch 1 activation?
3. Is Notch 1 activated by DLL1 or 4? Which cell is activating NOtch 1 in fibroblasts during diabetic healing? If DLL1 or 4 is blocked, either pharmacologically or genetically, does this downregulate Notch 1 in fibroblasts? This could be a potential therapy and could be addressed in the discussion.
4. What are IL6 levels in the Notch GOF murine wounds?

Reviewer #2 (Comments to the Authors (Required)):

The authors have shown increased Notch signaling in fibroblasts cultured from DFU compared with Non-diabetic human foot skin, and in wound fibroblasts of DFU and diabetic db/db and NOD mice compared with human skin and wounds from non-diabetic mice, respectively. They further analyzed the effects of Notch1 activation on wound healing in non-diabetic scenario. They demonstrated a delayed skin wound healing in genetically-modified mouse model which had activated Notch1 pathway in fibroblasts. Activated Notch1 pathway inhibited fibroblast proliferation, migration, and differentiation. Moreover, Notch1 activation in fibroblasts inhibited the angiogenic response of endothelial cells through inhibition of IL-6 expression. However, abovementioned changes were not found by knocking out Notch1 in fibroblasts.

In general, the questions proposed by this study are important, the results will bring advances to the field, the study is detailed and the data are mostly convincing. However, the comments below should be addressed before publication.

1. Although Notch1 activation in fibroblasts was demonstrated in diabetic wounds, the effects of Notch1 activation in fibroblasts for wound healing was studied in non-diabetic but genetically-modified model. To support the conclusion that Notch1 signaling determines fibroblast differentiation and fibroblasts-modulated angiogenic response in diabetic wounds, wound healing experiment should be performed in control and diabetic LOFCtrl and LOFNotch1 mice. If delayed wound healing in diabetic LOFCtrl mice could be improved in diabetic LOFNotch1 mice, their conclusion can be confirmed. Similarly, a difference may be seen in fibroblast migration, proliferation and differentiation between diabetic LOFCtrl and LOFNotch1 mice, since Notch signaling would be activated in fibroblasts from diabetic LOFCtrl mice but not in fibroblasts from diabetic LOFNotch1 mice.

2. It is interesting to see the difference in Notch signaling in fibroblasts from healthy skin and DFU. However, acute non-diabetic human wounds or chronic non-diabetic human wounds such as venous ulcers are better controls for DFU in order to understand the impact of diabetes on Notch pathway in wound fibroblasts. Is Notch pathway more activated in fibroblasts from DFU when compared with acute non-diabetic human wounds or chronic non-diabetic human wounds?

3. Hes-1 was analyzed to show the activation of Notch pathway in Fig. 1B and 1C. Although Hes-1 is one of Notch target genes (not specific for Notch1), its expression can be affected by other signaling pathways. The expression of activated Notch1 (Notch1 intracellular domain) should be analyzed to demonstrate the activation of Notch1 signalling pathway.
4. The method to isolate and verify primary fibroblasts should be briefly described although it has been described before in the cited paper. How were fibroblasts separated from keratinocytes, macrophages and endothelial cells from the tissue?
5. How specific is FSP-1 as a marker for fibroblasts? Is FSP-1 also positive in endothelial cells and keratinocytes? In the magnified image of C57BL6 wound (Figure 1), it seems endothelial cell in a blood vessel-like structure is also positively stained for FSP-1. In image of DFU in Figure 1, it seems all the cells are positively stained for FSP-1, since it is also green in epidermis.
6. Fig. 6 A-C: Expression of Notch1 intracellular domain should be shown as control for GFP/HDF and N1IC/HDF cells.
7. Basic clinical information of the human subjects should be presented, such as age, gender, HbA1c, medication, and diagnosis.
8. The quantification method of fluorescent images should be included in more details, and whether the signal intensity or percentage of positive cells or some other parameters are presented in the figure of quantification should be clear.
9. It states on The Jackson Laboratory website that Fsp1Cre (#012641) mice have BALB/c as background. Please confirm. What is the strain catalog number of NOD mice? Is it also BALB/c background?
10. It says on Page 20, line 451-452: Mouse skin and wound tissues obtained from 10-15 weeks old C57 BL6 (normal), and diabetic NOD (type I diabetes) and db/db (type I diabetes) mice. However, in line 456-457, it says 'NOD and db/db mice developed diabetes at 16~24 weeks old'. It is known that the median age for female NOD to develop diabetes is 18 weeks and male NOD mice have delayed onset of diabetes than females. Please clarify when the tissues were collected from the mice. How long the mice had been diabetic when they were wounded? Db/db mice are Type 2 diabetes model, and I think it is a typing mistake there.
11. The regulation of Notch signaling and the inhibitory effects of Notch1 activation on wound healing in diabetes have been shown before in keratinocytes, endothelial cells, fibroblasts, and macrophages (Front Immunol. 2017 Jun1;8:635; and PNAS April 2, 2019, 116 (14): 6985-6994). In discussion, the authors should discuss their results in relation to these previous findings regarding the role and regulation of Notch1 signaling in diabetic wound healing.

Minor points:

Figure 1: Denotation of NFF and DFUF should be included in figure legend.

Figure 1A: why Dll4 protein level was not analyzed? It is better to be included in order to have a general view of all the components of Notch signaling pathway.

Figure 2: statistical significance of wound healing should be tested using Two-way ANOVA with

repeated measures followed by post-hoc test.

Figure 7: what statistical analysis was used? It should be one-way ANOVA followed by post-hoc test.

We sincerely thank the reviewers for constructive criticisms and valuable comments, which were of great help in revising the manuscript. Accordingly, the revised manuscript has been improved with new data and additional interpretations. Our responses to the reviewer's comments are given below.

Reviewer #1:

In this paper the authors examine the role of Notch 1 in fibroblasts on angiogenesis, particularly in the setting of diabetic wounds. They identify that IL-6 is a functional Notch1 target and involved in regulating angiogenesis. The paper is well written and the experiments justify the conclusions- several novel mouse models were used as well as human diabetic wound tissues to increase the translational potential of these findings. The data are timely and very important in the field of diabetic wound healing. I have the following comments:

1. *For Figure 1C why was Day 7 chosen? Is there a change in Notch1 signaling any earlier? Does it persist in the diabetic setting? It would be interesting to see various time points and whether there is a sustained increase in the diabetic setting compared to control.*

Response: We thank the reviewer for the comment and fully agree with the reviewer. The reason why we were chosen day 7 is due to that it is a middle time point during the wound healing, and should very well reflect the Notch pathway activity in fibroblasts in wound tissues. We now examined additional two time points: day 1 (an early time point) and the last day when the wounds are almost healed (a late time point, which varies in different types of mouse), to assess dynamic changes of the Notch pathway activity in fibroblasts over the entire wound healing process. We tested levels of Hes-1 and N1^{IC} in fibroblasts to assess the Notch pathway activity, and observed that Notch pathway activation is "ON" in fibroblasts at Day 1, peaks in Day 7 and reduces to a very low level when the wounds are healed in diabetic mice (db/db and NOD), but remains undetectable in normal non-diabetic wounds (both non-ischemic acute wounds and ischemic chronic wounds in C57 BL6 mice) through wound healing process. These results not only confirm that the Notch pathway is activated or turned 'ON' in fibroblasts in murine diabetic wounds, while inactivated or turned 'OFF' in fibroblasts in non-diabetic murine wounds, but also show that the intracellular Notch pathway activation is dynamic in the fibroblasts throughout diabetic wound healing process. We now show these new immunostaining data as new *Suppl. Fig S5 and Fig S6* in the revised manuscript. We also updated the text accordingly.

2. *IN the Notch 1 GOF mice, does notch 1 activation in fibroblasts have any effect on the other cells (both structural and immune) - in other words, does the cellular milieu change in response to notch 1 activation?*

Response: We conducted histology analysis to examine the skin structure and immunostaining to examine lymphocytes (CD3⁺ T cells) in skin of the Notch 1 GOF mice. We did not find abnormal structure and morphology of skin cells except less collagen deposition in the skin of the Notch 1 GOF mice, which is consistent with impaired function of fibroblasts due to increased intracellular Notch1 pathway activity. Amounts of lymphocytes presented in the skin are almost undetectable in the Notch 1 GOF and control mice. These data suggest that no obvious changes in the cellular milieu in the skin of

Notch 1 GOF mice (except fibroblasts). We now show these new data as new *Suppl. Fig S8 and Fig S9* in the revised manuscript. We also updated the text accordingly.

3. *Is Notch 1 activated by DLL1 or 4? Which cell is activating Notch 1 in fibroblasts during diabetic healing? If DLL1 or 4 is blocked, either pharmacologically or genetically, does this downregulate Notch 1 in fibroblasts? This could be a potential therapy and could be addressed in the discussion.*

Response: We thank the reviewer for the comment. We have performed experiments to address this question. First of all, we have tested levels of Dll4 protein in NFF and DFUF and found that, like all other Notch ligands, Dll-4 is also increased in DFUF. We have updated this new data in Fig 1A. We have then used DAPT, a γ -secretase inhibitor, to suppress the pan-Notch pathway activation. We have also inhibited the Notch pathway activation by Jagged-1 neutralizing antibody (we tried to order Dll-4 neutralizing antibody, but it is on back-order and currently unavailable). We have found that both DAPT and Jagged-1 neutralizing antibody could inhibit the Notch pathway activation by reducing the levels of Notch 1 and Hey-1 in DFUF (mixture of 3 DFUF at 1:1:1 ratio) while Jagged-1 neutralizing antibody achieved a less extent inhibition compared to DAPT. These results suggest that intracellular Notch pathway activation observed in fibroblasts derived from DFU is dependent upon Notch receptor-ligand interaction. Likely, all ligands contribute to the Notch pathway activation, because all Notch ligands are upregulated in fibroblasts derived from DFUF and blocking of a single type of Notch ligand (by Jagged-1 neutralizing antibody) only partially suppress the Notch pathway activation. We now show these new data as new Fig 1B in the revised manuscript. The original Fig 1 B and 1C now become Fig 1C and 1D, respectively. We have also discussed the significance of targeting intracellular Notch pathway activity as a potential therapy for diabetic wounds. It is actually one of our future projects. The text in Results and Discussion has been updated accordingly.

4. *What are IL6 levels in the Notch GOF murine wounds?*

Response: The relative levels of IL-6 in the Notch GOF murine wounds were already shown in the original Fig 6D (it becomes new Fig 6E in the revised manuscript).

Reviewer #2:

The authors have shown increased Notch signaling in fibroblasts cultured from DFU compared with Non-diabetic human foot skin, and in wound fibroblasts of DFU and diabetic db/db and NOD mice compared with human skin and wounds from non-diabetic mice, respectively. They further analyzed the effects of Notch1 activation on wound healing in non-diabetic scenario. They demonstrated a delayed skin wound healing in genetically-modified mouse model which had activated Notch1 pathway in fibroblasts. Activated Notch1 pathway inhibited fibroblast proliferation, migration, and differentiation. Moreover, Notch1 activation in fibroblasts inhibited the angiogenic response of endothelial cells through inhibition of IL-6 expression. However, abovementioned changes were not found by knocking out Notch1 in fibroblasts.

In general, the questions proposed by this study are important, the results will bring advances to the field, the study is detailed and the data are mostly convincing. However, the comments below should be addressed before publication.

1. Although Notch1 activation in fibroblasts was demonstrated in diabetic wounds, the effects of Notch1 activation in fibroblasts for wound healing was studied in non-diabetic but genetically-modified model. To support the conclusion that Notch1 signaling determines fibroblast differentiation and fibroblasts-modulated angiogenic response in diabetic wounds, wound healing experiment should be performed in control and diabetic LOFCtrl and LOFNotch1 mice. If delayed wound healing in diabetic LOFCtrl mice could be improved in diabetic LOFNotch1 mice, their conclusion can be confirmed. Similarly, a difference may be seen in fibroblast migration, proliferation and differentiation between diabetic LOFCtrl and LOFNotch1 mice, since Notch signaling would be activated in fibroblasts from diabetic LOFCtrl mice but not in fibroblasts from diabetic LOFNotch1 mice.

Response: We thank the reviewer for the comment and fully agree with the reviewer. It was our original thoughts to test our hypothesis in LOF^{Notch1} and GOF^{Notch1} mice and their controls on diabetic background and submit our work to top-notch journals. However, it was very challenging and some of our efforts were not successful. For example, we tried to cross LOF^{Notch1} and GOF^{Notch1} mice with db/db heterozygous mice (db/db homozygous mice are infertile), but pups are not able to develop diabetes, due to the alteration of the original genetic background of db/db by BALB/cByJ background on FSP1-Cre mouse (FSP1-Cre mice were used for the generation of LOF^{Notch1} and GOF^{Notch1} mice). Unaltered original genetic background of db/db is absolutely required for offspring to develop diabetes. We are currently trying different breeding strategies to create LOF^{Notch1} and GOF^{Notch1} mice using other Cre mice (*i.e.* not FSP-1-Cre), which have close background with db/db mouse. Such efforts are undergoing and will take time. We hope that the reviewer can understand the reason and situation.

2. It is interesting to see the difference in Notch signaling in fibroblasts from healthy skin and DFU. However, acute non-diabetic human wounds or chronic non-diabetic human wounds such as venous ulcers are better controls for DFU in order to understand the impact of diabetes on Notch pathway in wound fibroblasts. Is Notch pathway more activated in fibroblasts from DFU when compared with acute non-diabetic human wounds or chronic non-diabetic human wounds?

Response: We thank the reviewer for the comment. Addressing the difference in the Notch signaling in fibroblasts from DFU and non-diabetic human wounds is what we have tried. Due to difficulty in obtaining tissues from acute non-diabetic human wounds, we, instead, employed acute non-diabetic mouse model to address such a question and showed our results in the original Fig 1C. We agree with the reviewer about using chronic non-diabetic human wounds as an additional control. However, our IRB protocol does not include tissues from human venous ulcers. It requires a new IRB protocol, which takes a few months to get approved, to do such a study. Even if with newly-approved IRB, it will be very difficult to recruit patients at this moment due to COVID-19 pandemic. We have thus tested mouse ischemic limb wounds, which is a type of non-diabetic chronic wounds and widely-used animal model to mimic human critical limb ischemic wounds. As shown in new Fig 1D (which was the original Fig. 1C) and new *Suppl. Fig S4*, the Notch signaling in fibroblasts at ischemic limb skin wounds is not activated at Day 7 and throughout entire wound healing process (new *Suppl. Fig S5 and Fig S6* at Day 1 and Day 13 when ischemic wounds are healed). The intracellular Notch pathway activation is dynamic

in the fibroblasts throughout diabetic wound healing process. We have updated these data in the revised manuscript.

3. Hes-1 was analyzed to show the activation of Notch pathway in Fig. 1B and 1C. Although Hes-1 is one of Notch target genes (not specific for Notch1), its expression can be affected by other signaling pathways. The expression of activated Notch1 (Notch1 intracellular domain) should be analyzed to demonstrate the activation of Notch1 signalling pathway.

Response: We have tested levels of N1^{IC} in mouse non-diabetic and diabetic wound tissues, and observed basically the same results as that of Hes-1 (results are updated in new *Suppl. Fig S6*). Similarly, we have tested levels of Hey-1, another Notch downstream target, in human DFU tissue samples, and observed similar results as that of Hes-1 shown in new Fig 1C (original Fig 1B) (note: due to the low quality of anti-human N1^{IC} antibody used for immunostaining, we instead tested Hey-1, another canonical Notch downstream target). We have shown this new data as new *Suppl. Fig S3*. Taken together, all these data are consistent and universally demonstrated that the Notch pathway activity in fibroblasts at diabetic wound tissues is activated compared to non-diabetic wounds. We hope that the reviewer can agree with us.

4. The method to isolate and verify primary fibroblasts should be briefly described although it has been described before in the cited paper. How were fibroblasts separated from keratinocytes, macrophages and endothelial cells from the tissue?

Response: We have added this information in the revised version of manuscript (page 22, line 16--page 23, line 4).

“Briefly, skin samples were treated in dispase (Roche) overnight at 4°C and then centrifuged to collect any released cells followed by removal of the epidermis (keratinocytes) from the dermis the following day based on established protocol (Normand and Karasek, 1995). Subsequently, the dermis was cut into small pieces, then treated with collagenase and hyaluronidase in DMEM-F12 (Invitrogen) for one hour at 37°C with stirring. The cell suspension was mixed with red blood cell lysis buffer, centrifuged, and then cells were collected and plated. Fibroblasts were grown in 1g/L glucose DMEM (Invitrogen), 10% FBS (HyClone), HEPES (Sigma-Aldrich) and Pen/Strep/Fung (Invitrogen), passaged after reaching confluence and second passage stocks were frozen in liquid nitrogen. Flow cytometry analysis was performed to confirm fibroblast identity based on positive staining for vimentin and CD-140a (PDGFR), and negative staining for CD31 and CD45.”

5. How specific is FSP-1 as a marker for fibroblasts? Is FSP-1 also positive in endothelial cells and keratinocytes? In the magnified image of C57BL6 wound (Figure 1), it seems endothelial cell in a blood vessel-like structure is also positively stained for FSP-1. In image of DFU in Figure 1, It seems all the cells are positively stained for FSP-1, since it is also green in epidermis.

Response: FSP-1 (Fibroblast-specific protein 1, also named as S100A4), originally thought to be fibroblast specific, is found to be also presented in a few other types of cells, including endothelial cells, according to Human Protein atlas (<https://www.proteinatlas.org/ENSG00000196154-S100A4/tissue>), although it is predominately expressed in fibroblasts. Hence, it is normal to see that some vessels are stained positive. In addition, keratin in skin tissues usually gives a high background in immunostaining,

and it is hard to eliminate such a background. Fibroblasts in skin wounds can be identified based on stronger FSP-1⁺ signals as well as their location in the granulation tissues and elongated morphological appearance. We hope that the reviewer can agree with us.

6. *Fig. 6 A-C: Expression of Notch1 intracellular domain should be shown as control for GFP/HDF and NIIC/HDF cells.*

Response: We have added expression of N1^{IC} data as new Fig 6A. The original Fig 6A-D now becomes Fig 6B-E.

7. *Basic clinical information of the human subjects should be presented, such as age, gender, HbA1c, medication, and diagnosis.*

Response: We have added *Suppl. Table 1* in the revised version of manuscript.

8. *The quantification method of fluorescent images should be included in more details, and whether the signal intensity or percentage of positive cells or some other parameters are presented in the figure of quantification should be clear.*

Response: We have added this information in the “materials and methods” in the revised version of manuscript (page 25, line 10-15).

“The relative levels of α -SMA/Hes-1/Hey-1/N1^{IC} expression were determined using ImageJ software. The fluorescence integrated intensity of Hes-1/Hey-1/N1^{IC} in each FSP-1 positive cells was measured and divided by its own FSP-1 integrated intensity. For α -SMA, its integrated intensity was divided by its nuclear DAPI intensity. Then each cell’s relative levels of α -SMA/Hes-1/Hey-1/N1^{IC} expression were grouped and average of relative fluorescence intensity was calculated.”

9. *It states on The Jackson Laboratory website that Fsp1Cre (#012641) mice have BALB/c as background. Please confirm. What is the strain catalog number of NOD mice? Is it also BALB/c background?*

Response: Yes, the Fsp1-Cre (#012641) mouse purchased from the Jackson Lab has BALB/c background. NOD strain purchased from the Jackson lab is NOD/ShiLtJ (#001976), which is a popular polygenic model for autoimmune type 1 diabetes, but not on BALB/c background. We have updated the strain catalog number of C57 BL6, NOD and db/db mice in the revised version of manuscript.

10. *It says on Page 20, line 451-452: Mouse skin and wound tissues obtained from 10-15 weeks old C57 BL6 (normal), and diabetic NOD (type I diabetes) and db/db (type I diabetes) mice. However, in line 456-457, it says 'NOD and db/db mice developed diabetes at 16~24 weeks old'. It is known that the median age for female NOD to develop diabetes is 18 weeks and male NOD mice have delayed onset of diabetes than females. Please clarify when the tissues were collected from the mice. How long the mice had been diabetic when they were wounded? Db/db mice are Type 2 diabetes model, and I think it is a typing mistake there.*

Response: We apologize for missing information about ages of NOD and db/db mice when the diabetic skin wound tissues were harvested, and a typo (db/db (type I diabetes)). Ages of 10-15 weeks old are C57 BL6 (normal and limb ischemic), while ages of NOD mice and db/db mice used in our study are 28-33 weeks old and 18-20 weeks old, respectively. We have updated the ages and sex information for all the mice used in our study. We also correct typo: (db/db (type II diabetes)) in the revised version of manuscript.

11. The regulation of Notch signaling and the inhibitory effects of Notch1 activation on wound healing in diabetes have been shown before in keratinocytes, endothelial cells, fibroblasts, and macrophages (Front Immunol. 2017 Jun1;8:635; and PNAS April 2, 2019, 116 (14): 6985-6994). In discussion, the authors should discuss their results in relation to these previous findings regarding the role and regulation of Notch1 signaling in diabetic wound healing.

Response: We have discussed it and cited these papers in the revised version of manuscript.

Minor points:

Figure 1: Denotation of NFF and DFUF should be included in figure legend.

Response: We have added this information in the revised version of manuscript.

Figure 1A: why Dll4 protein level was not analyzed? It is better to be included in order to have a general view of all the components of Notch signaling pathway.

Response: We have tested levels of Dll4 protein in NFF and DFUF and found that, like all other Notch ligands, Dll-4 is also increased in DFUF. We have updated this new data in Fig 1A. We did not include DLL-4 in the initial submission, because Dll-4 is known to be primarily expressed in endothelial cells. With the new data, we show that all the major Notch pathway components are upregulated in DFUF and demonstrate that the Notch pathway is activated in fibroblasts derived from DFU.

Figure 2: statistical significance of wound healing should be tested using Two-way ANOVA with repeated measures followed by post-hoc test.

Response: We have re-done ANOVA using two-way analysis. The results and statistical significance remain the same. Statistical results have been updated in the revised new Fig 2.

Figure 7: what statistical analysis was used? It should be one-way ANOVA followed by post-hoc test.

Response: Yes, it is ANOVA analysis. We have used one-way ANOVA for Fig 7 in the revised version of manuscript.

October 14, 2020

RE: Life Science Alliance Manuscript #LSA-2020-00769-TR

Dr. Zhao-Jun Liu
University of Miami
Molecular and Cellular Pharmacology
1600 NW 10th Ave
RMSB Bldg, RM1048
Miami, Florida 33136

Dear Dr. Liu,

Thank you for submitting your revised manuscript entitled "Notch1 Signaling Determines the Plasticity and Function of Fibroblasts in Diabetic Wounds". We would be happy to publish your paper in Life Science Alliance pending final revisions necessary to meet our formatting guidelines.

Along with the points listed below, please also address the following:

- please use the [10 author names, et al.] format in your references (i.e. limit the author names to the first 10)
- please add scale bars to Figure 3C, Fig. 4B, Fig. 5A,B

A. FINAL FILES:

-- Summary blurb (enter in submission system): A short text summarizing in a single sentence the study (max. 200 characters including spaces). This text is used in conjunction with the titles of papers, hence should be informative and complementary to the title. It should describe the context and significance of the findings for a general readership; it should be written in the present tense

and refer to the work in the third person. Author names should not be mentioned.

B. MANUSCRIPT ORGANIZATION AND FORMATTING:

Sincerely,

Shachi Bhatt, Ph.D.
Executive Editor
Life Science Alliance
<https://www.life-science-alliance.org/>
Tweet @SciBhatt @LSAjournal

Reviewer #1 (Comments to the Authors (Required)):

The authors have responded appropriately to my critiques.

Reviewer #2 (Comments to the Authors (Required)):

The authors were responsive to my concerns and presented a much-improved manuscript. The conclusions of the paper can be justified based on the presented data. I have no additional critiques.

October 16, 2020

RE: Life Science Alliance Manuscript #LSA-2020-00769-TRR

Dr. Zhao-Jun Liu
University of Miami
Molecular and Cellular Pharmacology
1600 NW 10th Ave
RMSB Bldg, RM1048
Miami, Florida 33136

Dear Dr. Liu,

Thank you for submitting your Research Article entitled "Notch1 Signaling Determines the Plasticity and Function of Fibroblasts in Diabetic Wounds". It is a pleasure to let you know that your manuscript is now accepted for publication in Life Science Alliance. Congratulations on this interesting work.

DISTRIBUTION OF MATERIALS:

Again, congratulations on a very nice paper. I hope you found the review process to be constructive and are pleased with how the manuscript was handled editorially. We look forward to future exciting submissions from your lab.

Sincerely,

Shachi Bhatt, Ph.D.

Executive Editor

Life Science Alliance

<https://www.life-science-alliance.org/>
